

# Emerging anthropogenic influence on Australian multi-year droughts with potential for historically unprecedented megadroughts

Georgina M. Falster[1,2]†, Nicky M. Wright[1,2]*†, Nerilie J. Abram[1,2,3], Anna M. Ukkola[4,5],
Benjamin J. Henley[6,7,8]

[1]Research School of Earth Sciences, Australian National University, ACT, 2601, Australia
[2]ARC Centre of Excellence for Climate Extremes, Australian National University, ACT, 2601, Australia
[3]Australian Centre for Excellence in Antarctic Science, Australian National University, ACT, 2601, Australia
[4]Climate Change Research Centre, UNSW Sydney, NSW 2052, Australia
[5]ARC Centre of Excellence for Climate Extremes, UNSW Sydney, NSW 2052, Australia
[6]School of Earth Sciences, University of Melbourne, VIC 3010, Australia
[7]School of Earth, Atmosphere and Environment, Monash University, VIC 3168, Australia
[8]ARC Centre of Excellence for Climate Extremes, University of Melbourne, VIC 3010, Australia
*Now at EarthByte Group, School of Geosciences, The University of Sydney, NSW 2006, Australia

†Note that GMF and NMW contributed equally to this manuscript.

*Correspondence to*: Georgina M. Falster (georgina.falster@anu.edu.au)

**Abstract**. In drought-prone Australia, multi-year droughts have detrimental impacts on both the natural environment and human societies. For responsible water management, we need a thorough understanding of the full range of variability in multi-year droughts and how this might change in a warming world. But research into the long-term frequency, persistence, and severity of Australian droughts is limited. This is partly due to the length of the observational record, which is short relative to the time scales of hydroclimatic variability, and hence may not capture the range of possible variability. Using simulations of Australian precipitation over the last millennium (850–2000 CE), we characterise the nature of multi-year meteorological droughts across Australia, and including a particular focus on the Murray-Darling Basin (MDB); the largest agricultural region in Australia. We find that simulated Australian droughts in the twentieth century (1900–2000) are within the bounds of pre-industrial natural variability in terms of drought intensity, severity, and frequency. A tendency to longer droughts in southwestern and eastern Australia (including the MDB) in the 20th century compared with the pre-industrial period suggests an emerging anthropogenic influence, consistent with projected rainfall changes in these regions. Large volcanic eruptions tend to promote drought-free intervals in the MDB. Model simulations suggest future droughts across Australia could be much longer than what has been experienced in the twentieth century, even without any human influence. With the addition of anthropogenic climate change—which favours drought conditions across much of southern Australia, due to reduced cool season rainfall—it is likely that future droughts in Australia will exceed historical experience.





## 1 Introduction

Droughts have severe environmental, societal, and economic impacts, which are expected to worsen in many mid-latitude regions with future anthropogenic climate change (Stevenson et al., 2022; van den Hurk R. Vautard K. von
Schuckmann S. Zaehle X. Zhang K. Zickfeld, 2021). Much of Australia is drought-prone, and drought risk is expected to increase in the future with ongoing human-caused climate warming (Kirono et al., 2020; Ukkola et al., 2020). Australia therefore requires careful water resource management and adaptation strategies. But despite the importance of understanding current and future hydroclimatic variability for developing water policy, research into the long-term frequency, severity, and typical duration of Australian droughts is limited (Kiem et al., 2016). This
is partly due to the length of the observational record of Australian precipitation (~120 years), which is short relative to the time scale of hydroclimatic variability (Stevenson et al., 2022; Morin, 2011). For example, Australia's Murray Darling Basin (MDB) has experienced four major multi-year droughts since national rainfall records began: the 1895–1903 Federation, 1935–1945 World War II, 1997–2009 Millennium and 2017–2019 Tinderbox droughts. Such multi-year droughts are infrequent enough in the observational record to make it challenging to acquire robust
statistics on drought characteristics such as maximum possible duration and severity, as well as return interval and overall time spent in drought. Palaeoclimate evidence further indicates that precipitation observations do not capture the full possible range of variability across the Australian continent (Freund et al., 2017; Ho et al., 2015; O'Donnell et al., 2021), potentially leading to underestimation of intrinsic drought risk in water policy (Leblanc et al., 2012; Vance et al., 2022).


Droughts lasting two or more years have particularly detrimental impacts on both the natural environment and human society. The latter effects are felt strongly in Australia's MDB, which incorporates Australia's largest river system and is the largest agricultural region on the continent. For example, impacts of the Millennium Drought included major losses in agricultural production (van Dijk et al., 2013; Leblanc et al., 2012), and catastrophic
ecological impacts which were exacerbated by continued water extraction from the MBD catchment (Semple et al., 2010; van Dijk et al., 2013; Leblanc et al., 2012; Zampatti et al., 2010; Paton et al., 2009). Major droughts in the MDB have typically been initially identified via their impacts to agriculture (Helman, 2009), although individual droughts have differed in their nature, severity, and possible causes (Verdon-Kidd and Kiem, 2009; O'Loingsigh et al., 2015). Impacts causing economic loss are also typically focussed on introduced/cultivated animal and plant
species (e.g., cattle and wheat), which are not adapted to the large natural hydroclimatic variability of Australia.





Precipitation across most of Australia has not experienced significant negative mean state trends during the observational record (Ukkola et al., 2019). In fact, the continent has experienced increasing rainfall in the area-mean, driven by a wetting trend in the northern parts of the country (Ukkola et al., 2019). However, a significant

and sustained decline in cool season rainfall has been observed in southwestern Australia and is projected to continue (Grose et al., 2020; Delworth and Zeng, 2014). Similar declining cool season rainfall trends also appear to be developing in the southeast, though with greater uncertainty across the ensemble of available future projections (Grose et al., 2020; BoM, 2022). Information from palaeoclimate records also suggests that droughts during the observational era are generally not beyond the range of variability of the past 500 years (Freund et al.,

2017; Palmer et al., 2015; Cook et al., 2016a; Ho et al., 2015; Vance et al., 2015), with the possible exception of parts of southern Australia ((Freund et al., 2017; Palmer et al., 2015; Cook et al., 2016a; Ho et al., 2015). However, palaeoclimate proxies containing information about pre-historical droughts are very sparsely distributed across the Australian continent, and this can lead to an incomplete assessment of drought risk (Ault et al., 2014). Nevertheless, climate models project that with continued global warming, Australia will dry on average (Stevenson et al., 2022),

and drought hazard metrics are predicted to increase significantly over the MDB (Kirono et al., 2020) and other southern parts of the country (Ukkola et al., 2020). This projected drying is particularly problematic given we do not fully understand the natural range of Australian hydroclimate extremes, and hence the degree to which intrinsic—as well as externally-forced—drought risk should be considered in future planning (Leblanc et al., 2012).


Here we explore the range of variability in multi-year droughts in Australia, with a particular focus on the MDB. We explore multi-year drought variability using a range of drought metrics (e.g., length, frequency, intensity, severity) and compare 20th century droughts in observations and simulations with simulated drought variability during the pre-industrial last millennium (850–1849). We seek to determine if (and if so, how) simulated droughts

have changed through time, including if droughts observed during the 20th century are more severe than expected from the simulated natural variability over the past millennium. We additionally assess whether MDB drought variability throughout the full last millennium (850–2000) is associated with any particular external forcing, for example anthropogenic greenhouse gas emissions.



## 2 Data and Methods

When assessing the nature of multi-year droughts, there are different ways to define drought intervals, particularly in terms of the hydroclimatic threshold that must be reached to classify a period as 'drought' (e.g. Cook et al., 2022; Mishra and Singh, 2010). In its Sixth Assessment Report, the Intergovernmental Panel on Climate Change defines drought as 'a period of abnormally dry weather that persists for long enough to cause a serious hydrological imbalance' (Douville et al., 2021). In this study we focus on meteorological droughts—which are characterised by

periods of below-normal precipitation—as we have long-term observations of Australian precipitation against which to compare climate model data. Additionally, climate models show higher skill for precipitation compared with other drought-relevant hydrological variables (Ukkola et al., 2018)

### 2.1 Observations

We used monthly gridded precipitation data from the Australian Water Availability Project (AWAP; (Jones et al.,

2009)), which is available at ~5 km resolution over Australia for 1900–present. We limited our analysis to the interval 1900–2000 since this is the interval covered by the climate model simulations we use (Sect. 2.2). We only assessed areas where primary observational data are available for >90% of the time series (Ukkola et al., 2019). In pixels with observational data for >90% of the time, any data gaps were filled using the monthly mean value. We converted monthly data to annual precipitation by summing monthly precipitation for calendar years (i.e., a

January–December total).

### 2.2 PMIP3/CMIP5 models

Long-term drought variability can be examined using multi-century climate model simulations. One of the most informative time periods for this is the pre-industrial last millennium (850–1849). With the exception of anthropogenic forcings, pre-industrial last millennium boundary conditions were very similar to those in the

modern period, allowing the pre-industrial last millennium to act as a 'baseline' against which we can compare modern (observed) and future (projected) climate variability. There is a specific Paleoclimate Modelling Intercomparison Project (PMIP) experiment targeting the last millennium interval ('past1000') (Schmidt et al., 2012, 2011). Climate models with past1000 experiments are generally too coarse to accurately simulate land-atmosphere interactions (Santanello et al., 2018; Müller et al., 2021), making precipitation amount one of the more

reliable indicators of drought variability in these simulations.



We assessed drought metrics using monthly outputs from 10 models (Table 1) submitted to the Paleoclimate Model Intercomparison Project/Coupled Model Intercomparison project with a past1000 experiment (PMIP3/CMIP5; Schmidt et al., 2012), as well as the Community Earth System Model Last Millennium Ensemble (CESM LME).

The CESM LME is the most comprehensive set of simulations covering the last millennium (Otto-Bliesner et al., 2016). The CESM LME has 13 ensemble members with all natural and anthropogenic forcings ('fully-forced'), as well as single-forcing experiments, with only one of the following external radiative forcings applied: well-mixed greenhouse gases, volcanic aerosols, orbital parameters, solar irradiance, and changes in land surface properties resulting from land use.


Using these models (PMIP3 and the CESM LME), we assessed whether the nature of multi-year droughts across Australia differ under pre-industrial versus 20th century forcing conditions. We focused on PMIP3 output rather than PMIP4 (Jungclaus et al., 2017) due to the greater availability of model simulations currently available. We additionally used the CESM LME to investigate the Murray-Darling Basin (MDB) multi-year drought response to

external forcings.

Model outputs for the past1000 and CMIP5 historical experiments span either 850–2000 or 850–2005 (Table 1), so we truncated data from all models to the year 2000. We used total precipitation (variable 'pr'), with native units $(kg/m^2/s)$ converted to millimetres per month by accounting for the number of days in each month. Annual

precipitation was determined by summing monthly precipitation for calendar years. We restricted the spatial extent to only grid cells over continental Australia and Tasmania.

### 2.2.1 Model verification

To assess how precipitation variability in the models compares with observations, we used two metrics: 1) the mean annual precipitation amount (MAP), and 2) the coefficient of variation (CV, where CV = standard deviation / mean)

for annual mean precipitation. For each metric, we compared the value in the models across 1900–2000 with observations over the same time period. We show the model results as absolute bias relative to observations. As the model resolutions differ from the resolution of the gridded observational dataset, we use a bilinear interpolation to regrid the observational dataset into each model's native resolution prior to assessing model performance. We quantified the similarity of the model values for the MAP and CV metrics with observational values using spatial

correlations.





## 2.3 Drought definitions and metrics

### 2.3.1 Definitions

There are many commonly-used methods for defining multi-year meteorological drought, and the specific definition used impacts drought metrics such as drought frequency and duration (Askarimarnani et al., 2021). In this study, we identified multi-year meteorological drought periods using the '2S2E' method (Coats et al., 2013), where a drought commences with two consecutive years of negative precipitation anomalies, and continues until two consecutive years of positive precipitation anomalies occur (grey shading in Fig. 1). This method allows for single 'wet' years during a drought. The 2S2E method has been commonly used in assessments of long-term multi-year drought variability in different regions (e.g. Coats et al., 2015; Cook et al., 2016b; Stevenson et al., 2015). Compared with other commonly-used methods of identifying multi-year droughts, 2S2E droughts tend to be shorter, but occur more frequently (Cook et al., 2022). Metrics of multi-year drought intensity are not significantly impacted by choice of drought definition (Cook et al., 2022).

### 2.3.2 Drought metrics

We characterised multi-year drought variability in observations and models using the following drought metrics: (i) length, (ii) relative intensity, (iii) relative severity, (iv) frequency, and (v) percentage of time spent in drought (Table 2). Drought length is the number of years in each drought, where the minimum drought length is two years. Drought intensity is the average yearly deviation from the climatology during drought events; due to differences between models, we show drought intensity as a relative precipitation deficit (%) rather than absolute deviation. For example, 0% represents the climatological mean precipitation, and 100% represents zero precipitation. Drought severity is the cumulative deviation from climatology during the drought, and is calculated by multiplying the relative intensity by length. For example, a value of 200% represents a total deficit equal to two years of mean precipitation. Drought frequency is the number of drought events during a given time period—for ease of comparison between the 20th century and last millennium, we express drought frequency as the average number of droughts per 100 years. Percentage of time spent in drought is the total number of years spent in drought per 100 years.

We calculated drought metrics on a per-model basis for a) their pre-industrial last millennium simulations (850–1849; piLM), and b) the 20th century (1900–2000) component of their historical simulations (HIST). We calculated the same metrics for the AWAP gridded observational product (1900–2000 only). For observations and the HIST





and piLM simulations, we used the 1900–2000 climatology for calculation of precipitation anomalies. Note that although the models' historical simulations extend back to 1850, we truncated these simulations to the 20th century to match the time period of the available observational data from AWAP.

In order to determine if any signals might have arisen without external forcing, we also calculated drought metrics

on each model's pre-industrial control simulation ('piControl'). For the piControl simulations, we used the long-term piControl mean as a climatological base period for calculation of precipitation anomalies. The piControl simulations are run for many years with external forcings fixed at pre-industrial values (i.e. those appropriate for the year 1850). As the length of models' piControl runs varies across the models (Table 1), we only analysed the last 500 years from each.


In the main text we report multi-model ensemble mean values for each drought metric for the HIST, piLM, and piControl simulations. Analyses were performed at the native resolution of each model, and regridded using a bilinear interpolation into 2° x 2° resolution to calculate multi-model ensemble means. When calculating multi-model ensemble mean values, to take into account the higher number of ensemble members in CESM LME, we

first took the mean value for all fully-forced CESM LME members, then incorporated the CESM LME mean into the multi-model ensemble mean. For easier visual comparison, in the main text we show observational data (AWAP) on the same 2° x 2° grid. To assess the skill of individual models in reproducing each drought metric, we calculated spatial correlations between observations and HIST simulations for the 1900–2000 period, using AWAP re-gridded to the same resolution of the model in question.

**2.3.2.1 Comparing drought metrics in the pre-industrial versus 20th century**

To assess whether the drought metrics were different under pre-industrial and industrial-era radiative forcing conditions, for each drought metric we subtracted the multi-model ensemble-mean piLM values from the multi-model ensemble-mean HIST values. To determine whether or not 20th century values are significantly different from the pre-industrial last millennium, we performed t-tests for each grid cell, comparing the distributions formed

by the individual model values for the piLM and HIST simulations.

**2.3.3 Drought metrics in the Murray-Darling Basin (MDB)**

To explore multi-year drought variability specifically in the MDB, we calculated the same drought metrics (Table 2) for MDB area-mean precipitation. To determine which models perform the best in simulating multi-year drought





characteristics in the MDB, for each metric we calculated the percent bias compared with observations in the 1900–
2000 period. We then summed the absolute percent bias values, and ranked the models from lowest to highest total
bias.

To assess the role of individual radiative forcings on MDB drought occurrence, we examined drought occurrence
in full- and single-forcing CESM LME members, and compared this with the radiative forcing associated with
solar, volcanic, and well-mixed greenhouse gas forcing (Schmidt et al., 2012). For this analysis, for each full- and
single-forcing sub-ensemble, we calculated the proportion of sub-ensemble members in drought in each year, and
compared this to the relevant forcings. For example, for the four-member volcanic-only single forcing sub-
ensemble, if no ensemble members are in drought in a particular year, then the proportion is zero. If all four
ensemble members are in drought, then the proportion is 1. If a particular forcing influences drought occurrence,
then we would expect changes in that forcing to coincide with periods of strong intra-ensemble agreement.

We further assessed the potential for external forcing of droughts in the MDB by examining the periods of highest
agreement across the entire 30-member CESM LME (all full- and single-forcing ensemble members). For this test
we aggregated the time periods when most (>67%) ensemble members were in drought at the same time, or most
(>60%) ensemble members were not in drought at the same time. The cut-offs for the two categories were the 10th
and 90th percentiles of all possible percentages of ensemble members in drought at the same time. We show the
distribution of values of external radiative forcings (well-mixed greenhouse gases, volcanic aerosols, solar
irradiance, and land use/land cover changes) applied to the models for each year in these two categories to test the
significance of external forcing in promoting drought conditions (or non-drought conditions) over the MDB.


We also assessed the influence of simulation length on the difference between MDB drought metrics in the pre-
industrial last millennium versus the 20th century. That is, to what degree does the fact that length of the piLM
simulations (1000 years) and HIST simulations (101 years) differ, affect any disparity between drought metrics in
the two time periods? To assess this, for each model, we randomly sampled 500 101-year segments from the 1000-
year-long LM simulations, to create distributions of possible values for each drought metric, all under past1000
forcing conditions. We then compared these distributions with the metrics calculated from the full LM simulations.





## 3 Results

### 3.1 Evaluation of climate models' precipitation variability

We first evaluated model skill in capturing observed precipitation variability at annual timescales. The observed
spatial mean annual precipitation (MAP) pattern is characterised by lower MAP in the continental interior
(particularly in the south), increasing toward the coast, and with maximum MAP at the north-eastern edge of the
continent and western Tasmania (Fig. 2a). Spatial variability in MAP from model simulations compares well with
observations in the 1900–2000 interval (mean spatial correlation (r) of 0.72). However, most models have a positive
overall MAP bias (Fig. 3); exceptions to this are CSIRO-Mk3l-1-2 and IPSL-CM5A-LR (negative overall bias).
Mean state bias aside, spatial MAP variability in IPSL-CM5A-LR is most similar to observations (r = 0.92), while
FGOALS-s2 is the least similar (r = 0.56). All spatial correlations are significant (p << 0.01). The spatial patterns
in MAP biases vary by model but they tend to overestimate MAP in the arid interior; any areas of underestimation
tend to be along the coast (Fig. 3).

The observed spatial coefficient of variation (CV) pattern is characterised by lower interannual precipitation
variability (CV ≈ 0.1–0.3) along the coastline (particularly in the south and east), increasing to higher CV values
inland where MAP is lower (Fig. 2b). All simulations show a broadly similar spatial pattern to observations (mean
r = 0.69), although most models underestimate the highest CV values observed in central Australia (Fig. 4). The
observed spatial precipitation CV patterns in the MDB are reproduced fairly well in the models (Fig. 4). Most
models have an overall negative CV bias, i.e., interannual precipitation variability is too low compared with MAP
(Fig. 4). Exceptions to this are the models with a negative MAP bias (CSIRO-Mk3l-1-2 and IPSL-CM5A-LR).
Spatial variability in precipitation CV in IPSL-CM5A-LR and MRI-CGCM3 is most similar to observations (r =
0.88), while the CESM-LME mean is the least similar (r = 0.48; individual LME members range 0.42–0.55). All
spatial correlations are significant (p << 0.01).


Together, MAP and CV provide insight into the models' ability to simulate annual precipitation variability. The
models' overall skill in simulating spatial and temporal precipitation variability relative to their own means suggest
that they provide a plausible representation of Australian precipitation, despite overall bias in mean MAP across
the continent.





### 3.2 Characteristics of droughts across Australia

### 3.2.1 Observed characteristics of Australian multi-year droughts

Observed mean multi-year drought length across 1900–2000 varies from 4–5 years along the eastern parts of Australia to over 10 years in parts of central and western Australia (Fig. 5a). The longest observed multi-year drought varies between ~8–20 years across Australia during the 20th century (Fig. 5e). In general, observed drought relative intensity and severity are lower along the northern, eastern, and southern Australian coastlines, then increase inland and toward the west coast (Fig. 6a,e). There is considerable spatial variation in observed drought frequency between 1900–2000, with 8+ multi-year droughts occurring in the MDB and across eastern Australia, and only ~5 multi-year drought events in parts of northern and central-western Australia (Fig. 7a). Although there are fewer discrete droughts per 100 years in central-western Australia, the overall proportion of time spent in drought is greater than in the south-east  (Fig. 7e), due to the greater mean drought length.

### 3.2.2 Characteristics of Australian multi-year droughts in model simulations of the 20th century

The 20th century simulations show less spatial variability in mean and maximum drought length (Fig. 5b,f) and frequency (Fig. 7b,f) than observed, with lower values particularly in inland areas. However, the low spatial variability in the multi-model mean largely arises from the averaging-out of the individual members which show a lack of consistent spatial patterns (Supp. Figs. 1-2, 5-6).  This is likely in part due to the high contribution of random variability to patterns produced in the short 101-year datasets. That is, the short time period hinders accurate characterisation of climatological patterns characterising multi-year droughts. This is also a likely contributor to the low spatial correlations between the models and observations across the 20th century (Supp. Figs. 1-6). While the spatial patterns vary across individual models, the drought length and frequency are of a similar magnitude to observations. The simulations also suggest similar spatial patterns and magnitudes to observations in drought intensity and severity.

### 3.2.3 Characteristics of Australian multi-year droughts in model simulations of the pre-industrial last millennium, and pre-industrial controls

Across all drought metrics, spatial patterns of variability in the piLM and piControl simulations resemble the structure that is evident in 20th century observations (Figs. 5-7). Individual model ensemble members also show spatial variability across all drought metrics in the pre-industrial last millennium (Supp. Figs. 7-12). In individual ensemble members, spatial patterns also generally have more structure in the piLM (Supp. Figs. 7-12) and piControl (Supp. Figs. 13-18) simulations than the HIST simulations. In the case of both the multi-mean means and the





individual simulations, this is likely because the greater length of the piLM and piControl simulations lessens the

contribution of random variability, and enhances climatic features.

The piLM and piControl simulations clearly demonstrate that Australian droughts can be much longer than what has been observed in the 20th century (Fig. 5g-h versus Fig. 5e-f). This finding is common across all individual ensemble members (Supp. Fig. 2 versus Supp. Fig. 8).


For other drought metrics, the realistic spatial patterns in the past1000 ensemble means mask notable differences in spatial patterns between individual ensemble members. Across the ensemble of models used here, IPSL-CM5A-LR and MPI-ESM-P are most commonly skillful in representing the spatial pattern of observed drought characteristics in Australia (r values in Supp. Figs. 1-6h,j). In both of these models, mean drought length has tended

to be longer in the 20th century than the pre-industrial last millennium (Supp. Fig 7h,j vs. Supp. Fig. 1h,j), whereas drought intensity is similar (Supp. Fig 9h,j vs. Supp. Fig. 3h,j). In IPSL-CM5A-LR (the most skillful on average), droughts are slightly more severe in the 20th century than the pre-industrial last millennium, particularly in central and western Australia (Supp. Fig. 4h vs. Supp. Fig. 10h). Drought frequency has minimal spatial variability in the piLM simulations of both models, suggesting this is a feature of the long-term climate system (Supp. Fig 11h,j).

However, the proportion of time spent in drought is generally longer in the 20th century than the pre-industrial millennium in both models, particularly southern and eastern Australia (Supp. Fig 6h,j vs. Supp. Fig. 12h,j).

### 3.3 Characteristics of droughts in the Murray-Darling Basin

Multi-year drought metrics were calculated from area-averaged annual precipitation over the MDB region to examine long-term drought characteristics for this region (Fig. 8). We analysed the results taking into account

model performance. Overall, BCC-CSM1-1 performs the best across MDB drought metrics for the twentieth century, with similarly good performance from HadCM3 and MRI-CGCM3 (Supp. Fig. 19). Total bias increases relatively sharply after MRI-CGCM3, with MIROC-ESM performing the worst overall for drought metrics in the MDB (Supp Fig. 19).

Mean observed multi-year drought length in the MDB was 6 years during the 20th century (Fig. 8). A similar value of 7.1 years is found for the ensemble mean of the HIST simulations (ranging 4–13.8 years for individual simulations). Mean MDB drought length is also similar for the longer simulations, with 5.8 years (range 3.7–7.4 years) in the piLM simulations, and 6.4 (5.1–8) years in the piControl. The longest observed drought in the MDB





lasted 12 years (Fig. 8), which is comparable to the HIST maximum drought length, with a multi-model ensemble
mean of 15.6 years (range 7–29 years); the mean for the three best-performing models is 11.7 years. Maximum
drought length in both the piControl runs (mean 26.6; range 19–37 years), and the piLM simulations (mean 22.4;
range 9–34 years) is much longer than the longest drought of the twentieth century, as was also evident in the
continent-spanning grids (Supp. Figs 8 and 14 vs. Supp. Fig. 2).

The relative intensity of 20th century droughts in the MDB is similar in observations and model simulations, with
a mean observed relative drought intensity of 16% and ensemble-mean simulated intensity of 13.8% (7–23%).
Model ensemble mean drought intensity in the piControl (mean 12.9; range 7–19) and piLM (mean 13.1, range 9–
18) simulations is also very similar to the twentieth century. Mean drought severity is also similar across
observations (78%), the HIST simulations (mean 79%; range 42–107%), piLM simulations (mean 67%; range 29–
122%), and piControl runs (mean 69%; range 42–106%). However, in the three best-performing models, drought
relative intensity and severity are on average worse in the 20th century than in the pre-industrial last millennium.

Both drought frequency and the total amount of time spent in drought are also similar across observations, and the
HIST, piLM, and piControl simulations. From observational data, the MDB experienced 7.9 multi-year droughts
per 100 years, and spent 48% of the time from 1900–2000 in drought. In model simulations of the same period, the
MDB experienced 8.4 (5–10.9) multi-year droughts per 100 years, and spent 54% (43–68%) of the time in drought.
In the LM simulations, the MDB experienced on average 8.2 (6.5–9.7) multi-year droughts per 100 years, and spent
48.8% (24–69%) of the time in drought. In the piControl runs, the MDB experienced on average 8.6 (7.5–10.6)
multi-year droughts per 100 years, and spent 54% (50–57%) of the time in drought.

### 345 3.3.2 Influence of difference in simulation length on drought metrics

During the 20th century, individual ensemble members show a large magnitude of spatial variability in drought
metrics over Australia, suggesting the 101 year sample length may not be fully representative of drought conditions
(Supp. Figs. 1-6). To examine this further, we resampled MDB area-mean precipitation in the piLM runs in 101-
year increments, and compared the distribution of values with the mean values for individual models' full piLM
simulations for each metric. Supporting Figure 20 demonstrates that the 101-year time period is not representative
of the full range of piLM drought variability. That is, individual 101-year segments do not capture the full range of
variability represented by the 1000-year piLM period. However, the results described in Sect. 3.3 are not majorly
affected by this bias. With one exception, the median value of the distributions formed by the 101-year segments



is very close to the mean value for the full piLM (blue dots compared with boxplots in Supp. Fig. 20). The exception
is maximum drought length, where few or none of the 101-year segments captured the longest possible drought of
the full pre-industrial last millennium (Supp. Fig. 20b).

Although the median values of the distributions of values from the 101-year segments mostly match the long-term
mean, there is large variability in the drought metrics across the shorter segments for most models (Supp. Fig.
20).This emphasises the value of longer simulations in determining the full range of possible drought conditions in
Australia's MDB. This is a particularly important consideration for assessing drought risk based only on
observational data, and for determining any anthropogenically forced changes in drought in observations and model
data that do not contain the full possible range of natural variability.

## 3.4 Possible anthropogenic influence on Australian multi-year droughts

We next assessed the potential role of anthropogenic climate change in influencing drought characteristics during
the 20th century. In eastern Australia and south-western Australia, droughts in the 20th century are on average
longer, and have a larger proportion of time spent in drought than in the pre-industrial last millennium (Fig 9a,f).
Specifically, in far south-western Australia, droughts are significantly longer on average in the 20th century than
the pre-industrial last millennium (stippling in Fig. 9a). Droughts also tend to be more intense, more severe, and
more frequently-occurring in the 20th century than the pre-industrial last millennium in this region (although these
differences are not significant). In parts of south-eastern Australia, droughts are on average longer in the 20th
century than the pre-industrial last millennium, with droughts also occurring more frequently, and a higher
proportion of time spent in drought (stippling in Fig. 9a,e-f). In all regions, the longest 20th century drought is still
much shorter than the longest drought in the pre-industrial last millennium (Fig. 9b).


Accordingly, the only instance in which the HIST simulations diverge significantly from the LM simulations across
a large area is for maximum possible drought length, where the longest droughts are uniformly much longer in the
piLM than HIST simulations (Fig. 9b). This is unlikely to be due to the addition of anthropogenic forcings, but
instead to the greater length of the piLM simulations which allow a more complete sampling of the full possible
range of variability (Sect. 3.3.2). There is no significant difference between drought intensity and severity in the
20th century versus the pre-industrial last millennium, in any part of Australia (Fig. 9c-d).





### 3.4.1 Forced droughts in the Murray-Darling Basin?

In the Community Earth System Model Last Millennium Ensemble (CESM LME), most external radiative forcings (Fig. 10a) are not associated with periods of MDB drought agreement in the corresponding ensemble members (Fig. 10c-d, f-h). The exception is volcanic forcing, where CESM LME most ensemble members run with volcanic forcing are not in drought after large eruptions (Fig. 10b-c, e). This is also clear in Supp. Fig. 21, where volcanic aerosol forcing is the only radiative forcing with a clear difference in values between periods when most CESM LME ensemble members are in drought, and when most are not in drought. Specifically, when volcanic forcing is large, a low proportion of ensemble members are in drought (Supp. Fig. 21b).

## 4 Discussion

Overall our results suggest that Australian droughts have not changed substantially in the last century compared to model simulations of the last millennium. In particular, drought intensity and severity observed during 1900-2000 were similar to those from the pre-industrial last millennium model simulations. However, the simulations suggest that multi-year droughts in parts of south-western and south-eastern Australia have been longer on average in the 20th century than the pre-industrial last millennium (Fig. 9a, f). Moreover, the maximum simulated drought length in long pre-industrial simulations is far longer than droughts Australia has experienced since commencement of widespread recording of monthly rainfall totals (Fig. 5e,f versus g,h). This is the case for both the forced (piLM) and unforced (piControl) simulations, suggesting that extremely long droughts of 20 years or more are a natural part of Australian hydroclimatic variability, and do not necessarily require anthropogenic forcing. If a drought of this length were to happen today it would have major societal and environmental implications beyond recent experience. Confidence in these results arises from evidence that the models used here simulate twentieth century Australian precipitation variability reasonably well, although in the ensemble mean have less spatial variability in multi-year drought metrics than is seen in observations. This is in part due to the fact that the 20th century is too short a time period for climatological patterns to emerge (Supp. Fig. 20), resulting in a high contribution of random variability in individual models (Supp. Figs. 1-6).

### 4.1 Influence of anthropogenic forcing on Australian multi-year droughts

Our findings demonstrate across a range of drought metrics that the instrumental period is not sufficiently long to experience the full possible range of natural variability in Australian multi-year droughts, particularly for drought





length (Supp. Fig 20). Hence, assessment of any anthropogenic contribution to modern multi-year droughts is difficult without the aid of model simulations and/or palaeoclimate proxy data.

The lack of strong spatial agreement in drought metrics between models in the piLM simulations—despite having similar radiative forcings applied (Table 1)—suggests that on the interannual to centennial time scales assessed here, Australian multi-year drought characteristics vary according to intrinsic variability rather than being associated with any particular external forcing (Supp Figs 7-12). For drought intensity, severity, and frequency, this dominance of intrinsic variability remains when considering the difference between the piLM and HIST simulations (Fig. 9c-e). The distinct spatial patterns in average drought length (Fig. 9a) and total time spent in drought (Fig. 9f) suggests that these differences are climatically meaningful, and hence may represent an emerging anthropogenic influence. Importantly, the regions spending on average more time in drought during the 20th century—eastern and south-western Australia—are also the parts of Australia where we expect to see human-caused rainfall declines during the 21st century (Ukkola et al., 2020; Cook et al., 2020). Hence, the lack of significant 20th century change across most of Australia in most drought metrics does not imply that there has been no human influence on Australian droughts during the 20th and 21st centuries, but rather that the 101-year HIST simulations are too short for significant differences to emerge. It is also noteworthy that the severe 2017-2019 eastern Australian drought was not included in this analysis.

## 4.2 Multi-year droughts in the Murray-Darling Basin

In the MBD, the difference between maximum drought length in the 20th century versus the pre-industrial last millennium is relatively small compared with the rest of the continent (Fig. 9b). Additionally, across the MDB the mean drought length is almost uniformly longer in the 20th century than the pre-industrial last millennium (Fig. 9a,f), suggesting a possible role for anthropogenic forcing in increasing both mean drought length, and the total proportion of time spent in drought. Major droughts in the MDB during the first two decades of the 21st century (i.e. the Millennium and 2017-2019 droughts) are not included in our analysis, but strengthen this finding of an apparent change towards longer droughts and more years spent in drought during the 20th century simulations compared with natural variability during the pre-industrial last millennium.

There is no obvious impact of anthropogenic forcing on drought occurrence in the MDB (Fig. 10a-d,h; Supp. Fig. 21a,d). However, large volcanic eruptions are associated with periods of less drought over the MDB (Fig. 10a-c,e; Supp. Fig. 21b). Volcanic eruptions generally trigger a transient 'El Niño-like' ocean-atmosphere response in





climate models (McGregor et al., 2020), including in the CESM LME (Stevenson et al., 2016). This is unlikely to
drive periods of increased rainfall over the MDB (Tozer et al., 2023), and hence this response requires further
investigation.

**4.3 Comparison with palaeoclimate proxy evidence**

Our findings support the available evidence from the sparsely-distributed proxy records for Australian hydroclimate
throughout the last millennium, which similarly suggest that observational-era droughts are not exceptional in the
context of the past few hundred years. Hydroclimatic reconstructions for south-western Australia (F-MAM-JJA-
SO rainfall reconstruction for 1350-2017 CE; O'Donnell et al., 2021), eastern Australia (DJF Palmer Drought
Severity Index reconstruction for 1500-2012 CE; Cook et al., 2016a; Vance et al., 2015), the MDB (annual rainfall
reconstruction for -749-1980 CE; Ho et al., 2015), and south-eastern Queensland (annual rainfall reconstruction
for 1000-2012 CE; Kiem et al., n.d.) almost uniformly suggest that observational-era droughts are not unusual in a
multi-centennial context. In fact, each of these reconstructions suggests that droughts in their respective regions
have in the past been longer and/or more severe than those experienced in the late 19th and 20th centuries. The
only exception is Freund et al. (2017), who concluded that the cool-season precipitation deficit during the
Millennium Drought in south-eastern Australia *was* highly unusual in the context of the past 400 years. A possible
explanation for this discrepancy is that our analyses only examined annual-mean precipitation, whereas observed
eastern Australian rainfall declines of the past decades have been focussed in the cool season (Pepler et al., 2021;
Speer et al., 2021). However, Freund et al.'s reconstruction also does not extend far enough back to intersect earlier
dry intervals inferred from other reconstructions (Ho et al., 2015; Cook et al., 2016b).

**5 Conclusions**

In PMIP3/CMIP5 models and the CESM LME, multi-year droughts in parts of south-western and eastern Australia
(including the MDB) have been longer on average in the 20th century than the pre-industrial last millennium, with
a larger proportion of time overall spent in drought. Anthropogenic forcing is the likely cause of this difference.
Conversely, droughts in inland western Australia are slightly shorter than in the pre-industrial last millennium.

Throughout the last millennium, large volcanic eruptions have been associated with relatively drought-free
intervals in the Murray-Darling Basin. However, although we assessed the possible role of external forcing on
Australian multi-year droughts, we did not investigate the role internal ocean-atmosphere variability plays in



predisposing various regions of Australia to multi-year drought. This will also be a topic of future study and of particular interest as several major modes of variability, including ENSO and the Indian Ocean Dipole, are projected to intensify in the future (van den Hurk et al., 2021).


Finally, the longest droughts of the 1000-year-long pre-industrial last millennium simulations are much longer than the longest droughts of the twentieth century. This suggests that very long (20+ years) 'mega-droughts' are a natural feature of Australian hydroclimate, but that the historical period has not yet been long enough to record one of these mega-droughts. Given droughts are likely to intensify with future anthropogenic forcings (Kirono et al.,

2020), water management authorities should plan for significantly longer, as well as more intense, droughts than have been experienced in the twentieth century.

**Data availability**

All climate model data used in this study are freely available from online repositories. PMIP3/CMIP5 simulations are available from Earth System Grid Federation nodes https://esgf.llnl.gov/index.html. The CESM1 Last Millennium Ensemble is available

from the Earth System Grid https://www.earthsystemgrid.org/. The version of AWAP used in this study is available from AMU on request.

**Contributions**

GMF and NMW contributed equally to the paper and are listed in alphabetical order. GMF and NMW performed the analyses and made the figures, with input from AMU, BJH, and NJA. GMF and NMW wrote the paper, and all

authors contributed to editing and review.

**Competing interests**

The authors declare that they have no conflict of interest.

**Acknowledgements**

We acknowledge support from the Australian Research Council Centre of Excellence for Climate Extremes

(CE170100023). This research was made possible by computational resources provided by the Australian National Computational Infrastructure (NCI), including resources awarded through the NCI and Australian National



University merit allocation schemes. AMU acknowledges support from an ARC Discovery Early Career Researcher Award (DE200100086). Thanks to Professor Michael Roderick for initial discussions leading to the conception of this paper.

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

**Table and figures**

| Model | Atmospheric resolution (lon x lat) | Time span (*past1000* and historical) (CE) | Full length of piControl (years) | Transient radiative forcings | | |
|---|---|---|---|---|---|---|
| | | | | solar | volcanic | LULC |
| BCC-CSM-1-1 | 128 × 64 | 850–2000 | 500 | VSK | GRA | piControl |
| CCSM4 | 288 x 192 | 850–2005 | 1051 | VSK | GRA | PEA |
| CSIRO-Mk3L-1-2 | 64 × 56 | 850–2000 | 1000 | SBF | CEA | piControl |
| FGOALS-s2 | 128 x 108 | 850–2005 | 501 | VSK | GRA | piControl |
| GISS-E2-R | 144 × 90 | 850–2005 | 5525 | VSK | GRA | PEA |
| HadCM3 | 96 x 73 | 850–2005 | 1201 | SBF | CEA | PEA |
| IPSL-CM5A-LR | 96 × 96 | 850–2005 | 1000 | VK | GRA | piControl |
| MIROC-ESM | 128 × 64 | 850–2005 | 630 | DB | CEA | piControl |
| MPI-ESM-P | 196 × 98 | 850–2005 | 1157 | DB | CEA | PEA |
| MRI-CGCM3 | 320 x 160 | 850–2005 | 500 | DB | GRA | piControl |
| CESM LME | 144 x 96 | 850–2005 | 1156 | VSK | GRA | PEA |

**Table 1. Details of PMIP3 models used in this study, including references for the transient radiative forcings applied to the pre-industrial last millennium (850-1849) simulations. LULC = land use/land cover, VSK = solar forcing from (Vieira et al., 2011), DB = solar forcing from (Delaygue and Bard, 2011), SBF = solar forcing from (Steinhilber et al., 2009), GRA = volcanic forcing from**
**(Gao et al., 2008), CEA = volcanic forcing from (Crowley et al., 2008), PEA = LULC forcing from (Pongratz et al., 2008). We used the GISS-E2-R past1000 experiment denoted '28'. The CESM LME comprises 13 simulations with all natural and anthropogenic forcing, as well as greenhouse gas only (n=3), volcanic-only (n=4), orbital-only (n=3), solar-only (n=4), and LULC-only (n=3) single-forcing simulations.**




| Drought metric name | Drought metric definition |
| --- | --- |
| Length | Number of years in a drought, with a minimum of two |
| Relative intensity | Average yearly deviation from the precipitation climatology during drought events |
| Relative severity | Cumulative deviation from the precipitation climatology across the drought, i.e., relative intensity multiplied by length |
| Frequency | Number of droughts occurring per 100 years |
| Percentage of time spent in drought | Number of years spent in drought per 100 years |


**Table 2. Summary of drought metrics.**








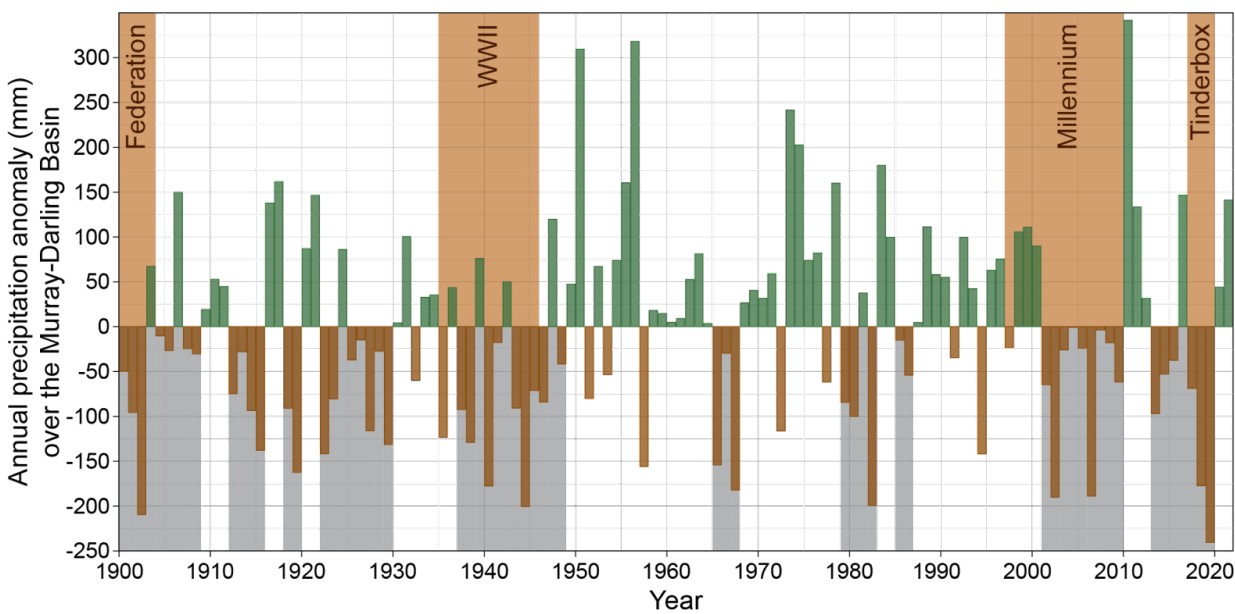


**Fig. 1. Area-averaged annual precipitation anomalies over the Murray-Darling Basin (MDB) for 1900–2022. Anomalies are relative to the entire interval. Orange shading shows historical MDB droughts mentioned in the text. Grey shading shows droughts identified by the '2S2E' method (Sect. 2.3.1).**

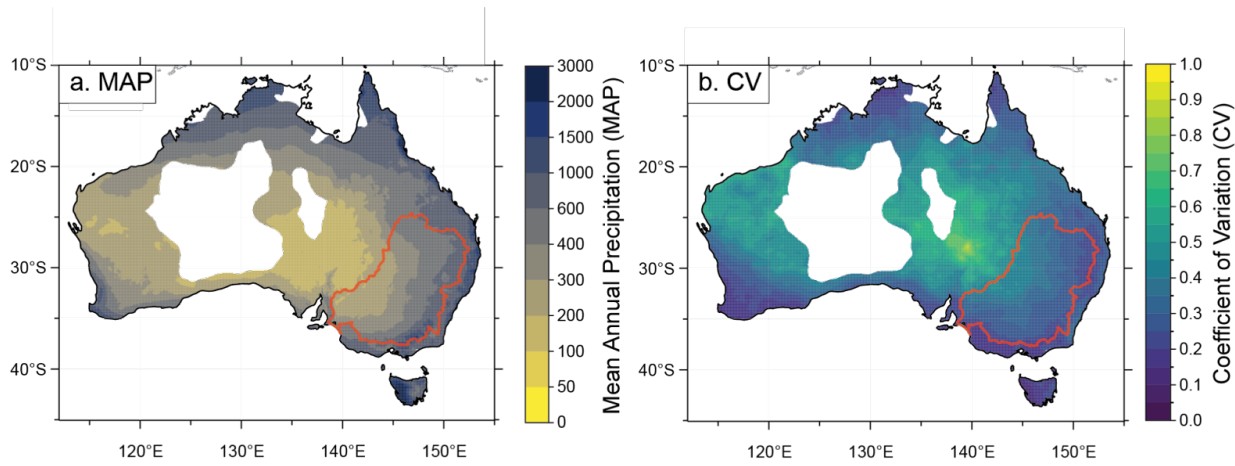


**Fig. 2. Summary of spatial characteristics of Australian rainfall, in terms of a) Mean Annual Precipitation (MAP; mm/year), and b) Coefficient of variation (CV). Data are from AWAP, showing mean values for the 20th century (1900-2000). The Murray-Darling Basin is outlined in red.**




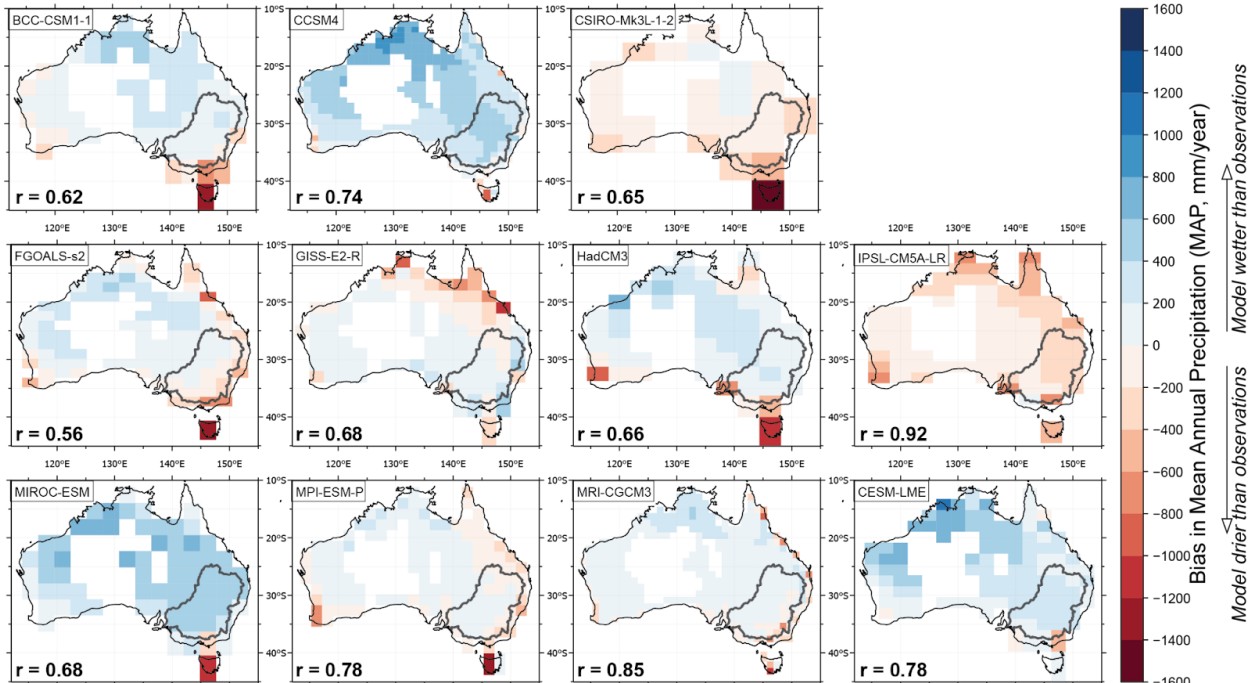

**Fig. 3 Mean annual precipitation (MAP) maps across PMIP3 simulations considered in this study (all panels except bottom right), and CESM-LME ensemble mean (bottom right panel) for the 20th century (1900–2000). The Murray-Darling Basin is outlined in black. Maps show bias (in mm/year) relative to AWAP. Spatial correlations of each model with AWAP are shown in the bottom left corner. All correlations are significant (p < 0.05).**



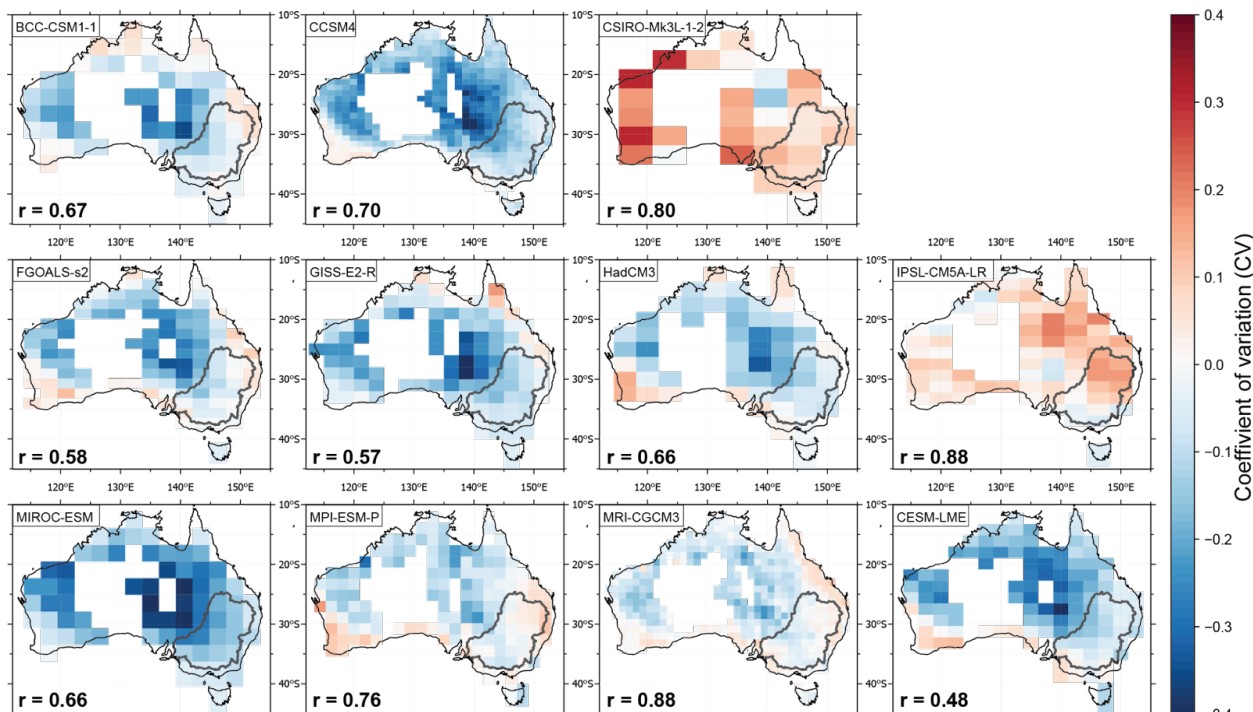


**Fig. 4 Coefficient of variation (CV) maps across PMIP3 simulations considered in this study (all panels except bottom right), and CESM-LME (bottom right panel) for the 20th century (1900–2000). The Murray-Darling Basin is outlined in black. Maps show bias relative to AWAP. Spatial correlations of each model with AWAP are shown in the bottom left corner. All correlations are significant (p < 0.05).**


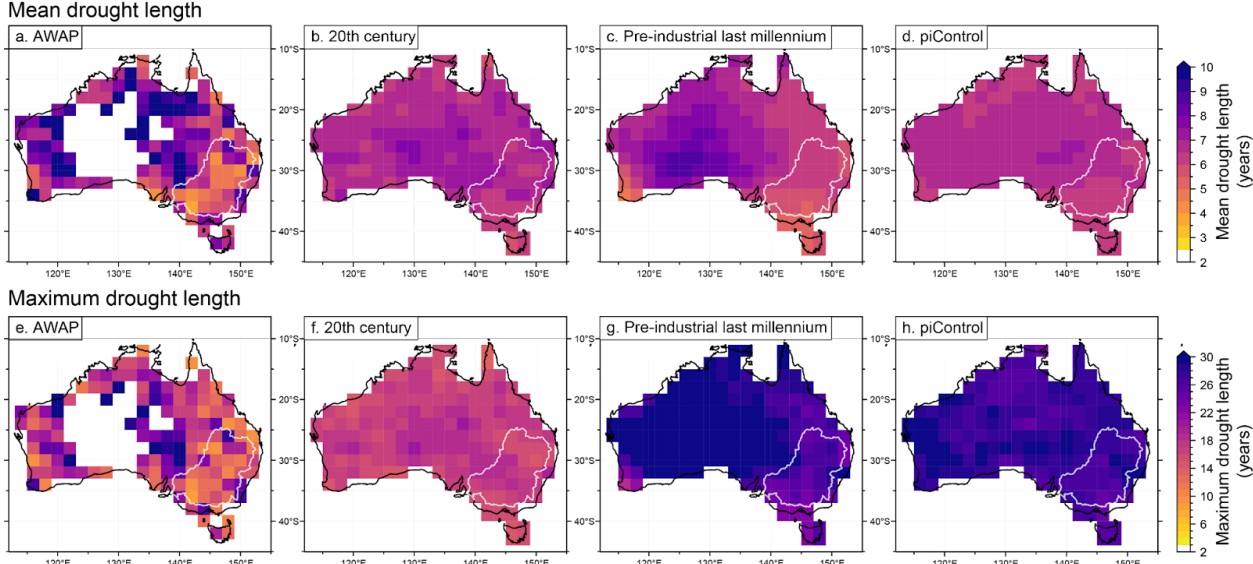

**Fig. 5. Summary of mean (a–d) and maximum (e–h) multi-year drought length in observations (a, e); and multi-model ensemble means in the 20th century (1900-2000) (b, f), pre-industrial last millennium (850-1849) (c, g), and piControl simulations (d, h). White outline shows the Murray-Darling Basin.**



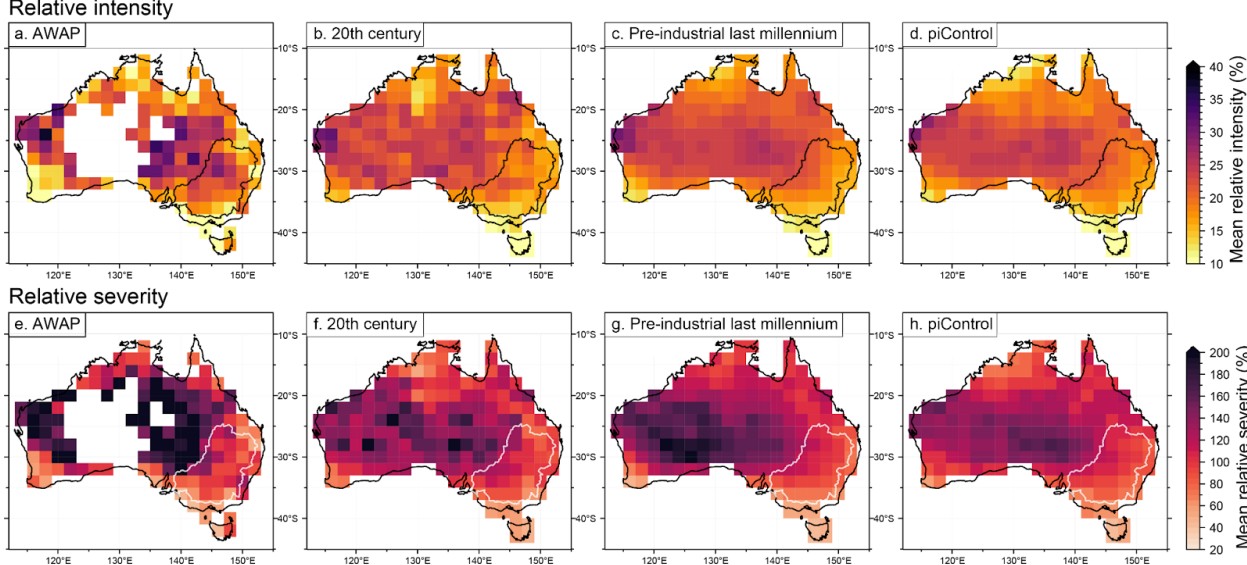

**Fig. 6. Relative intensity (a–d) and severity (e–h) of multi-year droughts in observations (a, e); and multi-model ensemble means in the 20th century (1900-2000) (b, f), pre-industrial last millennium (850-1849) (c, g), and piControl simulations (d, h). Relative intensity is reported relative to the individual models' 1900-2000 precipitation mean. Relative severity is relative intensity multiplied by drought length. Outline shows the Murray-Darling Basin.**

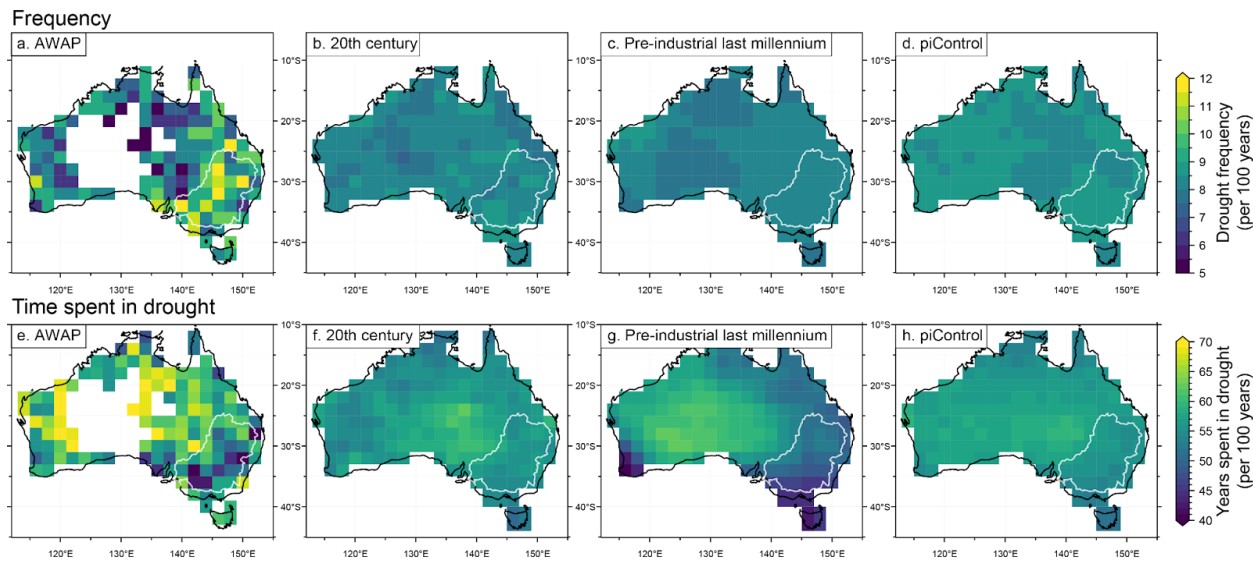

**Fig. 7. Frequency (a–d) and number of years spent in drought (e–h) per 100 years in observations (a, e); and multi-model ensemble means in the 20th century (1900-2000) (b, f), pre-industrial last millennium (850-1849) (c, g), and piControl simulations (d, h). White outline shows the Murray-Darling Basin.**



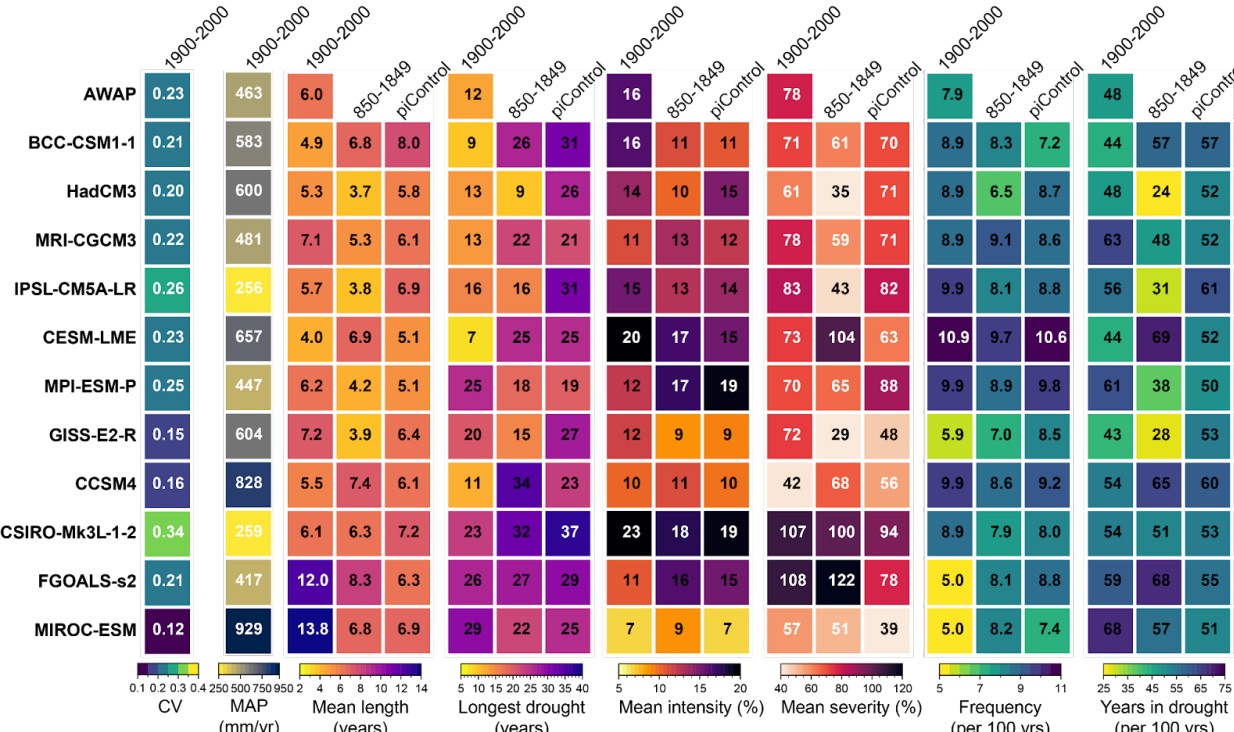

**Fig. 8. Observed and modelled drought metrics for the Murray Darling Basin.** Summary of drought metrics for observations, individual PMIP3/CMIP5 simulations, and the CESM LME mean, using area-averaged annual precipitation in the MDB. For each drought metric (aside from CV and MAP, which are used only to assess model skill), the first column shows metrics for the 20th century (1900-2000), the middle column shows metrics for the pre-industrial last millennium (850–1849), and the final column shows metrics for the piControl simulations. Models are ranked according to their summed percent bias across all drought metrics in the twentieth century (ordered best to worst; Supp. Fig. 19).





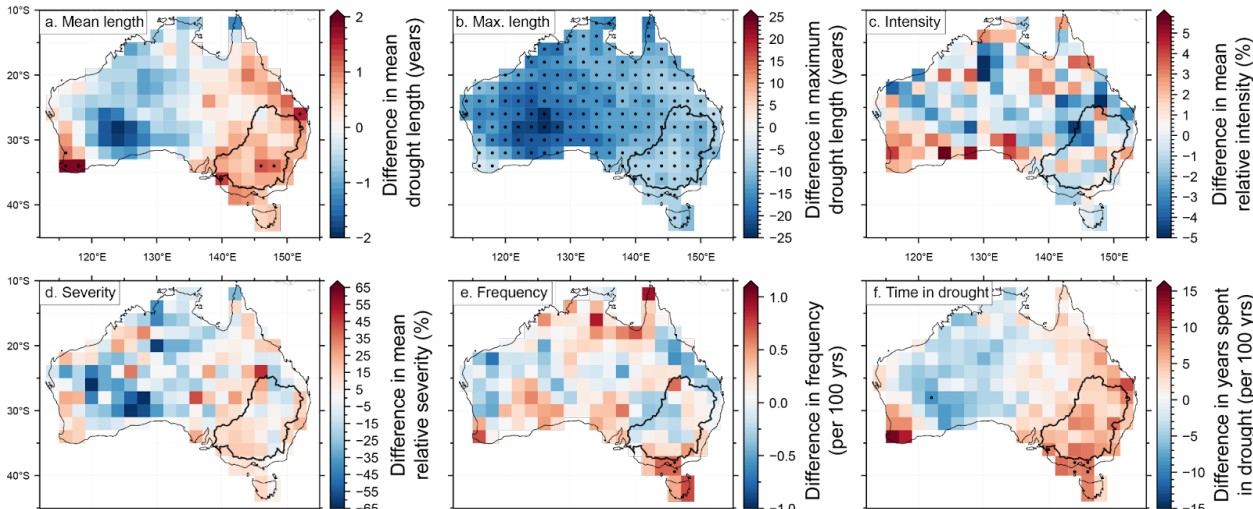

**Fig. 9 Difference between the 20th century (1900-2000) and pre-industrial last millennium (850-1849) multi-model ensemble means for a) mean drought length, b) maximum drought length, c) drought intensity, d) drought severity, e) drought frequency (droughts per 100 years), and f) years spent in drought per 100 years. Reds denote that the drought metric is 'worse' (longer, more intense/severe, more frequent, or more years are spent in drought) in the 20th century than the pre-industrial last millennium. Blues denote that the drought metric is worse in the pre-industrial last millennium than the 20th century. Stippling denotes that the 20th century multi-model mean value is significantly different to the pre-industrial value ($p < 0.05$). Murray-Darling Basin is outlined in black.**



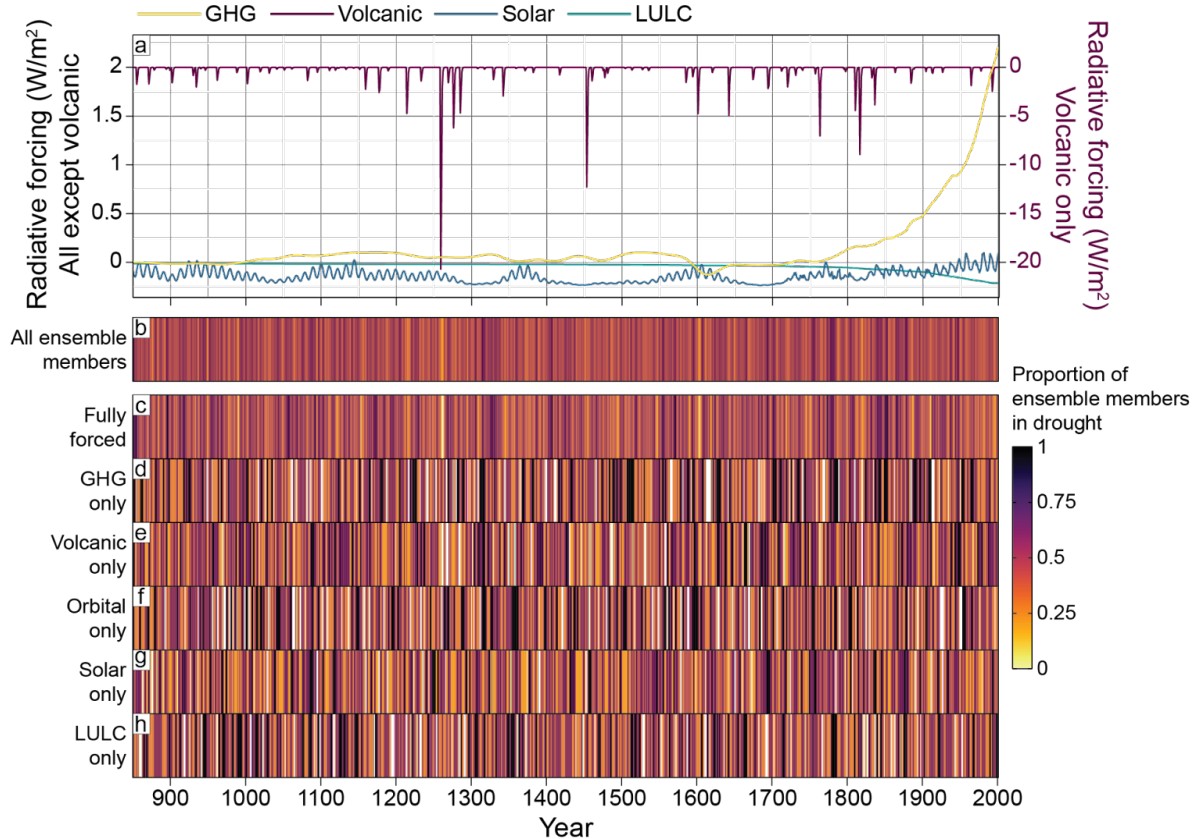

**Fig. 10. Occurrence of multi-year droughts in the Murray-Darling Basin, in each member of the CESM LME. a) Radiative forcings applied to the CESM LME (Schmidt et al., 2012). b) Proportion of all CESM LME ensemble members (n=30) in drought throughout the full last millennium (850-2000). c) Proportion of fully-forced CESM LME ensemble members (n=13) in drought. d) Proportion of GHG-only CESM LME ensemble members (n=3) in drought. e) Proportion of volcanic-only CESM LME ensemble members (n=4) in drought. f) Proportion of orbital-only CESM LME ensemble members (n=3) in drought. g) Proportion of solar-only CESM LME ensemble members (n=4) in drought. h) Proportion of LULC-only CESM LME ensemble members (n=3) in drought. In panels b-h), colours correspond to the proportion of full- or single-forcing sub-ensemble members that are in drought in each year. GHG = greenhouse gas. LULC = land use/land cover.**