# Peer review of "Potential for historically unprecedented Australian droughts from natural variability and climate change"

_EGUsphere, 2023_

## Referee Comment (RC1)

The authors have presented a well-motivated and, largely, clearly executed analysis of changes in large droughts in Australia using validated modelled drought outputs for the historical period and over the last millennium as derived from paleoclimate data. I believe that this manuscript is a worthy contribution to the scientific discourse, but I believe some revisions and additional analysis is required to make this a robust study as follows:

- My most pressing concern is ensuring that the validation is sound and appropriately quantified;
- At present, I do not believe that the presentation of figures intended for communicating the impact of different sample sizes is sufficient; and
- I feel that a measure of drought intensity that is comparable across events of different length is missing

More detail for these three dot points are provided under the general comments.

And finally, I would like to see an acknowledgement of the limitations in making comparisons with historical droughts, particularly since the analysis omits most of the millennium and all of the Tinderbox droughts.

Otherwise, there are a small number of clarifications, particularly of the caveats, and these are detailed in the minor comments.

**General comments**

A brief discussion is needed to acknowledge the limitation of using one definition of a water year for all of Australia and locations where the water year definition used is most/least relevant.

I need to see more detail about how the spatial correlations were calculated. Was this a calculation of correlations between matching locations that were then averaged across the region, or was the significance at each location assessed independently, or was a field significance considered and if so, which method was used (e.g. false discovery rate, walker's test, counting test)? The latter (a measure of field significance) is what is required. An averaging of correlation results is not appropriate, and assuming spatial independence is also inappropriate. The results of quantifying the field significance of the similarities between the observed and modelled droughts will impact on the credibility and interpretability of the remaining results. If fiend significance results markedly alter the validity of the modelled results, the interpretation of the pre-industrial millennia results will need to be re-interpreted accordingly.

The measures of both relative drought intensity and severity appear to be functions of the average deviation from climatology across the event. It appears to me that relative drought severity is a superfluous metric since drought length is also presented (although the figures show the mean over time and sometimes across models, so I realise it is showing something different then taking the product of, say, fig 5b and fig 6b). What seems to be missing is a measure that reflects the most severe drought year (or years) such as the most intense two consecutive years of drought within an event and to see if the maximum (annual or consecutive multi-annual) intensity is changing in different time periods. A measure like this would prevent the metric from being influenced by the definition of the event duration,

which is the case for drought intensity and severity metrics, particularly since event length would be sensitive to the definition of drought in determining onset and termination.

Section 3.2.3. Results of Fig 6 b-d and f-h are presented in the text, but references to the figures need to be made. I recognise that reference is made to supp. Figs 1-2 and 5-6 that reflect the points made in the text in more detail, but Fig 6 b-d and f-h are still relevant and need to be referenced in the text.

The specification of significance level needs to be stated in terms of what significance level has been chosen for the evaluation i.e. $\alpha = 0.01$, rather than reporting overall p-values.

Section 3.3. I would like some text around how the precipitation mean and variance compare between modelled and observed specifically in MDB (rather than relying on the reader to interpret the figures themselves) as this could help explain the difference in drought results.

Section 3.3.2. and supporting fig 20. It does not follow that large spatial variability implies anything about the adequacy of record length. Another justification is needed here. Also, I can see the intent of what the authors are aiming to communicate here: that shorter simulations fail to fully explore how variable drought can be given the range of drought conditions that can be explained over a longer period. I think Supp Fig 20 b is sufficient because it is clear that the maximum drought length obtained from a shorter 101-year sample will likely underestimate the maximum plausible drought length. However, I don't think the remaining plots in this figure demonstrate that the drought characteristics sampled in a 101 year-long sample are not representative of what could be reasonably expected in our climate and an alternative way of presenting this data is needed.
One suggestion I have would be to plot a cumulative density line or scatter for each of the 500 samples on a single plot and then overlayed would be the cumulative density line for the 1000-year long simulation. As a dummy example, I've done this for 500 samples of n=100 for a normal distribution of mean=10 and sd=2 (these values have no meaning or significance, it's just for an example) with an additional sample of n=1000 shown in bold. The historical or HIST ecdf could also be added and discussed with respect to over/underestimating flood characteristics at different magnitudes with respect to the longer record.

[Figure]

I'd be more than happy for the authors to either adopt this or develop an alternative for presenting their findings that would provide a figure that supports the argument they are making in the text of section 3.3.2.

Minor comments:
L48: A reference is needed for the Tinderbox drought being a "major" drought.
L102: Full stop at the end of this sentence
L105: Use "0.05° × 0.05° latitude/longitude resolution" for consistency with later model resolution descriptions.
L146: clarify that the bias relative to observations is shown for each member as well as the overall ensemble mean.
L193 and elsewhere: specify that the resolution is for latitude/longitude
L209: is the percentage bias also reported for the rest of Australia? If not, why not?
L222: could you clarify in L126 how many members are run in natural or fully forced or single forcing so it is clear what this "30" is based on?
L225: to improve clarity, extend this sentence with "...(>60%) ensemble members were not in drought at the same time *as this would indicate*....."
L233: replace "," with "...(101 years) differ *and* affect any disparity...."
L235: for clarity, it would be worth reconfirming the number of distributions that are generated (i.e. 500)
L255: do you mean "observed variability" instead of "MAP"?
L264: in addition to overall bias in mean MAP across the continent, the models also largely generate precipitation with reduced variability (with the exception of CSIRO-Mk31-1-2 and IPSL as previously stated).
L280: to improve clarity, insert "across ensemble members" prior to referencing the supp figs.
L285: does the statement "suggest similar spatial patterns" apply to all members, or just some? If this is across the ensemble, please state this up front in the paragraph as this would help demonstrate that the simulations are an adequate representation of the observations, which I believe is the intent of this paragraph, and it is key to providing a basis on which comparisons of HIST, piLM, and pi Control can be assessed. The message at present is a little

lost because the shortcoming are presented first and the purpose of the 20th century simulations is not clearly stated (i.e. for validating the model runs and providing credibility for the piLM and pi control runs).

L311: particularly *in* southern and eastern Australia

L323: for consistency, include "ranging from" or "range of" prior to "5.1-8"

L325: Is this supposed to be "mean maximum drought length" for both metrics on this line?

L328: I'm not sure what is meant by "continent-spanning grids". Is this just "all locations"? Or "grids covering the mainland"?

L330: could you clarify which model simulations? I believe it would be the HIST model simulations

L332: add "%" to these numbers

L335-336: Does this statement not apply across the ensemble results too? Or is it just confined to the three best performing models?

L336: "worse" is subjective. Use "more severe" or similar

L385: "The exception is volcanic forcing, where  most ensemble members *in the CESM LME* run with volcanic forcing are not in drought….". Also, it seems like a discussion of the agreement between ensemble members under LULC forcing is missing. It would also be good to comment on the variability of the forcing as it's very easy to see when volcanic forcing imposes a large change in the radiative forcing, but the variations in solar and LULC are less easy to identify.

Line 391: specify that this is in reference to results that are averaged across Australia.

L415 to 417: Given the findings that 100-year samples result in different summary statistics compared to a single 1000 year record, these comparisons really should be made in the context of the distribution of 100 year samples taken from the longer record as opposed to comparing a 100 year long record with a 1000 year long record (i.e. fig 9a-d).

L427: MDB (rather than MBD)

L438: Can text be added to make this finding a bit more explicit? Such as: "The co-occurrence of volcanic eruptions and supressed drought conditions over the MDB appear to contradict existing understandings of the impacts of volcanic eruptions on El Niño-like conditions and subsequent impacts on rainfall in the MDB".

L 443-L445 needs to be clarified. At present it appears to contradict the first sentence of the conclusion.

---

## Referee Comment (RC2)

Review on **Emerging anthropogenic influence on Australian multi-year droughts with potential for historically unprecedented megadroughts**

The study focuses on deepening the understanding of the natural ranges of Australian droughts, which can be used to better assess intrinsic and externally forced drought risks in future planning. They address this by comparing droughts in the 20th century (1900-2000) based on observations and models, with simulated droughts during the pre-industrial millennium (850-1849). They seek to assess if drought characteristics (mean duration, maximum length, intensity, etc.) have changed during the last century, compared to the past millennium.

One of the main conclusions is that multi-year droughts have been longer on average during the 20th century over part of Australia (including the MDB), compared to the last millennium, and that anthropogenic forcing is the likely cause of this change.

The authors also conclude that having a larger sample (pre-industrial millennium) allows for a better characterization of natural drought variability, reaching out extreme events (longest droughts) that have not been observed over the last century. Based on this, the authors conclude that such extreme events are part of the natural range of droughts in Australia, and thus they can be expected in the future. This, superimposed with projected drying trends, pose critical challenges for adaptation planning.

The article is well written and the motivation and research question is clear. However, there are some methodological aspects that should be addressed before drawing robust conclusions:

1) There are models that perform better than others during the historical period, and this is quantified as part of the analysis (Sect. 2.2.1; 2.3.3; Supporting Fig. 19). In this line, the interpretation of results should also account for these different performances. We should trust more those models that better represent the observations in the historical period, right?

   For example, if the physical mechanisms represented by a particular model structure leads to lower interannual precipitation variability compared to observations, it is expected that its simulations during the last millennium reflect the same bias, and vice versa for the case of higher interannual variability. However, the conclusions of the paper are based on the average of all models, independently of their performances during the historical period.

   Given the large differences between models (spatial precipitation patterns, MAP values and performance against observations), I don't think mean ensemble values can be directly used for interpretation of results. For example, the assessment of the "Possible anthropogenic influence on Australian multi-year droughts" (Section 3.4) relies on the ensemble mean of models, however, we know that there are models that perform better than others.

A way to account for these different model performances could be to apply a statistical correction to the models before analyzing droughts, similarly than those applied to GCMs in the historical period before analyzing their future projections (e.g., Cannon, 2018 and references therein). This data-process involves that each model is corrected according to their own performances in the historical period, and then results can be interpreted similarly across models.

From Sect. 2.3.2, it is inferred that droughts are defined as deviations from the climatology of each model (right?) If the models are bias corrected, the same climatological mean (that from AWAP) could be used for drought definition. And direct comparison between models could be applied, instead of %. This is easier for interpretation than "For example, 0% represents the climatological mean precipitation, and 100% represents zero precipitation". Same for severity, it would be much easier to compare directly mm across models, instead of % ("For example, a value of 200% represents a total deficit equal to two years of mean precipitation.")

Comparing deviation metrics as % is influenced by the native MAP of each model (deviations from a low absolute MAP values represent larger % than when the MAP is larger). By comparing Fig 2.a and Fig. 3, it can be seen that some models have MAP biases up to 100%, with similar absolute biases that observed MAP in Fig. 2a.

*Cannon, A.J. Multivariate quantile mapping bias correction: an N-dimensional probability density function transform for climate model simulations of multiple variables. Clim Dyn**50**, 31–49 (2018). https://doi.org/10.1007/s00382-017-3580-6*

2) Having a more extreme event in a large sample can be somehow expected, but I am missing an assessment of the return period of such events. The longest droughts simulated over the pre-industrial millennium, can be expected to happen over the next century, couple of centuries, thousand years?

In the same line, I think that for providing evidence for adaptation planning, the longest droughts should be assessed in conjunction with their deficits: it is not the same to communicate that 20-years of minor droughts (e.g., 0-10% deficits) can be expected that to communicate that 20-years of severe droughts (e.g., >40% deficits) can be expected in the future. This could be done by accounting for relative severity together with maximum length.

Minors comments:
Supp. Fig. 7: "Mean multi-year drought length in (a) observations (1900-2000) and (b-l) model simulations of the pre-industrial last millennium (850-1849). Showing the CESM LME ensemble mean." It should say, panel Fig. 7l presents the CESM LME ensemble mean. Same for all figures.

Title: it is a complicated title that I don't think is communicating the main messages of the paper. I recommend the authors to consider a simpler one.

Abstract: "Model simulations suggest future droughts across Australia could be much longer than what has been experienced in the twentieth century, even without any human influence." This can be misunderstood as future projections, please re-phrase. An option could be: Drought simulations over the last millennium suggests that future droughts across Australia could be much longer than what has been experienced in the twentieth century, even without any human influence.

---

## Author Comment (AC1)

**Author response:** We thank the Reviewer for their kind comments, and for their constructive review of our manuscript. We have addressed all suggestions (details below). In response to both these comments, and the suggestions of Reviewer 2, we will make the following changes to improve the clarity of our manuscript:

- Add two new Supporting Figures:
    - One showing the return period of multi-year droughts in the Murray-Darling Basin, in each PMIP3 model's pre-industrial last millennium simulation
    - One showing the relative severity of the longest droughts in each simulation from each model, as well as observations
- Provide increased clarity around the interpretation of Supporting Figure 20, and also slightly modify this figure for ease of interpretation (details below)
- Provide more detail of calculation of the spatial correlations, and - at the Editor's discretion - either use these correlations to weight the calculation of multi-model means, or add a statement as to why we did not do this
- Add a statement as to why we did not bias-correct the models

We also provide two new analyses in this response:

- A comparison of multi-year drought characteristics in the observational dataset used in this paper with observations that extend to 2021, thereby encompassing the Millennium and Tinderbox droughts.
- For each model, a timeseries of the *maximum* relative intensity of each drought across 850-2000 CE for the MDB, based on the Reviewer's suggestion.

Additionally, we will address all general and specific comments from the Reviewer as outlined below. We consider that these changes will result in a stronger paper with clearer, more robust findings.

**Review of 'Emerging anthropogenic influence on Australian multi-year droughts with potential for historically unprecedented megadroughts'**

The authors have presented a well-motivated and, largely, clearly executed analysis of changes in large droughts in Australia using validated modelled drought outputs for the historical period and over the last millennium as derived from paleoclimate data. I believe that this manuscript is a worthy contribution to the scientific discourse, but I believe some revisions and additional analysis is required to make this a robust study as follows:

- My most pressing concern is ensuring that the validation is sound and appropriately quantified;
- At present, I do not believe that the presentation of figures intended for communicating the impact of different sample sizes is sufficient; and
- I feel that a measure of drought intensity that is comparable across events of different length is missing
- More detail for these three dot points are provided under the general comments.

- And finally, I would like to see an acknowledgement of the limitations in making comparisons with historical droughts, particularly since the analysis omits most of the millennium and all of the Tinderbox droughts.

*Author response: Regarding the last dot point, which is not addressed under 'General comments': we re-calculated all multi-year drought metrics, but using a version of the observational dataset that extends to the year 2021 (thereby including the Millennium and Tinderbox droughts). We provide that comparison on the following page. For mean and maximum drought length, relative severity and intensity, and proportion of time spent in drought, differences between the two datasets are negligible. Drought frequency increases slightly with the addition of the extra 21 years - particularly in eastern Australia.*

*We currently state in the Discussion: "Major droughts in the MDB during the first two decades of the 21st century (i.e. the Millennium and 2017-2019 droughts) are not included in our analysis, but strengthen this finding of an apparent change towards longer droughts and more years spent in drought during the 20th century simulations compared with natural variability during the pre-industrial last millennium" (L431). We will re-phrase this as follows, to state that our conclusions—particularly regarding anthropogenic impacts on Australian droughts—may be influenced by the fact that the PMIP3 simulations do not span the Millennium or Tinderbox droughts.*

"We note that the PMIP3 *past1000* simulations do not cover two of the Murray-Darling Basin's most impactful droughts of the historical period: the Millennium and 2017-2019 droughts. However, the occurrence of two major droughts in the first two decades of the 21st century provides additional support for our finding that the MDB is spending more time in drought during the historical period compared with natural variability during the pre-industrial last millennium."

[Figure]

[Figure]

Otherwise, there are a small number of clarifications, particularly of the caveats, and these are detailed in the minor comments.

**General comments**
A brief discussion is needed to acknowledge the limitation of using one definition of a water year for all of Australia and locations where the water year definition used is most/least relevant.

*Author response: We will add the following brief description of possible implications of using a January-December year across the broad range of climate zones in Australia, at Line 110.*

"Precipitation seasonality varies across Australia. For example, in tropical northern Australia, 'years' may be better represented by 'tropical years', where each year starts e.g. in May of calendar year 1 and finishes in April of calendar year 2. However, this is not applicable across the entire continent, where some regions have winter-dominated precipitation, and others have no distinct seasonality. Given this study focuses on multi-year events, our choice of a calendar year for calculation of annual totals should not have a major influence on results."

I need to see more detail about how the spatial correlations were calculated. Was this a calculation of correlations between matching locations that were then averaged across the region, or was the significance at each location assessed independently, or was a field significance considered and if so, which method was used (e.g. false discovery rate, walker's test, counting test)? The latter (a measure of field significance) is what is required. An averaging of correlation results is not appropriate, and assuming spatial independence is also inappropriate. The results of quantifying the field significance of the similarities between the observed and modelled droughts will impact on the credibility and interpretability of the remaining results. If fiend significance results markedly alter the validity of the modelled results, the interpretation of the pre-industrial millennia results will need to be re-interpreted accordingly.

*Author response: For Supp. Figs. 1-6, we show a pattern correlation. We will add the following, more detailed, description to the Methods (at Line 150): "Spatial correlations between pairs of two-dimensional grids were calculated by flattening each grid, resulting in two directly-comparable vectors, with each index position of each vector representing the values at a particular latitude-longitude pair. We calculated the Pearson correlation coefficient, and provide an estimate of the significance of that coefficient (reported 'significant' if p < 0.05)."*

*For Fig. 9 where we calculate significance per pixel, we did not account for spatial dependence or False Discovery Rate. We will adjust the p-values as suggested by the Reviewer, and update our findings if necessary.*

The measures of both relative drought intensity and severity appear to be functions of the average deviation from climatology across the event. It appears to me that relative drought severity is a superfluous metric since drought length is also presented (although the figures show the mean over time and sometimes across models, so I realise it is showing something different then taking the product of, say, fig 5b and fig 6b).

*Author response: That is correct - the 'drought severity' metric combines the information from the drought length and drought intensity metrics; drought intensity is independent from duration. The severity metric provides an estimate of the* total *precipitation deficit over the drought, which is not easily apparent from inspecting the length and intensity metrics separately. We consider this to be an important aspect of the drought characterisation processes, as it provides insight into the total water stress placed on the region over the span of a particular drought. We also chose this particular metric of drought severity as it is consistent with our other metrics (e.g. drought mean and maximum length), which are expressed for the whole event.*

What seems to be missing is a measure that reflects the most severe drought year (or years) such as the most intense two consecutive years of drought within an event and to see if the maximum (annual or consecutive multi-annual) intensity is changing in different time periods. A measure like this would prevent the metric from being influenced by the definition of the event duration, which is the case for drought intensity and severity metrics, particularly since event length would be sensitive to the definition of drought in determining onset and termination.

*Drought severity is indeed dependent on the metric used to identify droughts. In the '2S2E' method used in this paper, multi-year droughts tend to be longer and more severe (but less intense) compared with, for example, a method where 'multi-year droughts' are two or more years below the 20th percentile of climatological precipitation. Given the already very large number of figures in this paper, we do not provide a test of the sensitivity of our results to different drought identification methods. However, we will add a statement to this effect.*

*Whilst we agree that the new analysis proposed by the Reviewer is certainly an interesting one, assessing temporal variability in maximum drought intensity is not practical for a continent-wide analysis. Here we have performed this analysis for the Murray-Darling Basin: the below figure shows, for each model, the* maximum *relative intensity of each drought, for 850-2000 - i.e. the single most intense year within the drought (as proposed by the Reviewer). Across the PMIP3 models, there is no consistent temporal variability in drought maximum intensity through time.*

[Figure]

Section 3.2.3. Results of Fig 6 b-d and f-h are presented in the text, but references to the figures need to be made. I recognise that reference is made to supp. Figs 1-2 and 5-6 that reflect the points made in the text in more detail, but Fig 6 b-d and f-h are still relevant and need to be referenced in the text.

***Author response:*** *We will make the following changes to ensure specific reference to Fig 6b-d and f-h:*
- *At Line 286, we will reference Fig. 6b and f at the end of the line.*
- *At Line 290, we will edit the figure reference to state 'Figs. 5-7a,e compared with Figs 5-7c,d,g,h'. We will also add the word 'broadly' before 'resemble' in Line 289*

The specification of significance level needs to be stated in terms of what significance level has been chosen for the evaluation i.e. = 0.01, rather than reporting overall p-values.

***Author response:*** *We will state that we used α = 0.05 as our significance threshold.*

Section 3.3. I would like some text around how the precipitation mean and variance compare between modelled and observed specifically in MDB (rather than relying on the reader to interpret the figures themselves) as this could help explain the difference in drought results.

*Author response: We did not originally include this text, as the models' skill in simulation precipitation variability over the MDB is very similar to the skill over the entire continent. We will add the following text to a new sub-section of Section 3.1. Accordingly, we will remove the current sentence about CV in the MDB at Line L254.*

"**Section 3.1.1 Evaluation of climate models' precipitation variability: Murray-Darling Basin**
Model skill in capturing observed precipitation variability over the Murray-Darling Basin (MDB) is very similar to model skill over the entire continent. Most models have a positive overall MAP bias (Fig. 3); exceptions to this are CSIRO-Mk3l-1-2 and IPSL-CM5A-LR (negative overall bias). Although most models slightly underestimate precipitation CV in the MDB, the spatial CV patterns are reproduced fairly well (Fig. 4)."

Section 3.3.2. and supporting fig 20. It does not follow that large spatial variability implies anything about the adequacy of record length. Another justification is needed here.

*Author response: Major and random spatial variability, rather than well-defined spatial patterns, implies a large random element to the patterns. In this particular context, this suggests that there are insufficient multi-year droughts for a climatological pattern to emerge. This points to an important role of model internal variability at these time scales as the different ensemble members discussed here only differ in their internal variability (rather than other factors such as model physics). To clarify, we will add the following text at Line 348:*

"That is, the presence of random spatial variability, rather than well-defined spatial patterns, implies a large random element. In this context, this suggests that there are insufficient multi-year droughts in a 101-year sample for a climatological pattern to emerge."

Also, I can see the intent of what the authors are aiming to communicate here: that shorter simulations fail to fully explore how variable drought can be given the range of drought conditions that can be explained over a longer period. I think Supp Fig 20 b is sufficient because it is clear that the maximum drought length obtained from a shorter 101-year sample will likely underestimate the maximum plausible drought length. However, I don't think the remaining plots in this figure demonstrate that the drought characteristics sampled in a 101 year-long sample are not representative of what could be reasonably expected in our climate and an alternative way of presenting this data is needed.

*Author response: Supporting Figure 20 demonstrates that no single 101-year segment can capture the full range of variability present in a 1000-year sample. Hence, any single 101-year segment (such as the historical period) likely gives a skewed representation of long-term variability in that particular drought metric. This is true for all metrics, not just maximum possible drought length.*

*We recognise that by showing the overall mean values from the piLM simulations as blue dots on Supp. Fig. 20, we are not making this point particularly well. We will therefore replace these with dots showing the equivalent values from each model's **HIST** simulation. We will also add the following text to Section 3.3.2 to clarify this (replacing the current sentence at Line 351-352):*

"That is, individual 101-year segments do not capture the full range of variability represented by the 1000-year piLM period, with the magnitude of drought durations, intensity, severity, frequency, and proportion of time spent in drought varying markedly from one 101-year period to another. This means that selecting any single 101-year

period (such as the historical period) is not representative of the full variability in the models' simulated precipitation."

One suggestion I have would be to plot a cumulative density line or scatter for each of the 500 samples on a single plot and then overlayed would be the cumulative density line for the 1000-year long simulation. As a dummy example, I've done this for 500 samples of n=100 for a normal distribution of mean=10 and sd=2 (these values have no meaning or significance, it's just for an example) with an additional sample of n=1000 shown in bold. The historical or HIST ecdf could also be added and discussed with respect to over/underestimating flood characteristics at different magnitudes with respect to the longer record. I'd be more than happy for the authors to either adopt this or develop an alternative for presenting their findings that would provide a figure that supports the argument they are making in the text of section 3.3.2.

*Author response: Thank you for the suggestion. In response to this and a similar suggestion from Reviewer 2, we will add a new supporting figure showing the return period of droughts of different lengths in the MDB, from each model's piLM run. However, showing return periods calculated from the 101-year segments on that same plot is slightly misleading. For example, the longest single drought of the piLM simulation will occur only once in the piLM run, giving it a return period of ~1000 years, while in the 101-year segment containing that same drought, it will have a return period of ~101 years. This makes it hard to compare the two analyses on the sample plot and may be misinterpreted by readers. This type of analysis is also slightly complicated by the fact that multi-year droughts are discrete events that do not occur every year.*

*In replacing the blue spots showing values for the full piLM simulations with spots showing values for the HIST simulations (as detailed above), we consider that we will more clearly demonstrate that using a too-short time period leads to an inaccurate understanding of multi-year drought characteristics. Combined with our additional text as shown above, this will better support our arguments in section 3.3.2.*

**Minor comments:**
L48: A reference is needed for the Tinderbox drought being a "major" drought.
*Author response: We will add the following reference: Nguyen, H., M. C. Wheeler, H. H. Hendon, E.-P. Lim, J. A. Otkin, The 2019 flash droughts in subtropical eastern Australia and their association with large-scale climate drivers, Weather and Climate Extremes, Volume 32, 2021. https://doi.org/10.1016/j.wace.2021.100321.*

L102: Full stop at the end of this sentence
*Author response: Thank you for catching this. We will add this full stop.*

L105: Use "0.05° × 0.05° latitude/longitude resolution" for consistency with later model resolution descriptions.
*Author response: We will make the suggested edit.*

L146: clarify that the bias relative to observations is shown for each member as well as the overall ensemble mean.
*Author response: We will edit the sentence to "For each model, we show the results as absolute bias relative to observations."*

L193 and elsewhere: specify that the resolution is for latitude/longitude
*Author response: We will edit this and similar sentences to "...into 2° x 2° (lat x lon) resolution…"*

L209: is the percentage bias also reported for the rest of Australia? If not, why not?
*Author response: We only calculate the percent bias values for the Murray-Darling Basin. This is because the MDB analyses are performed on a single area-mean precipitation timeseries (rather than each individual grid*

*cell), which permits these more detailed analyses. We feel this is less appropriate if calculated on area-mean precipitation over the whole continent due to the many climate zones this would encompass.*

L222: could you clarify in L126 how many members are run in natural or fully forced or single forcing so it is clear what this "30" is based on?
*Author response: We will add this information to Lines 127-129 in the following format: "...well-mixed greenhouse gases (n=3), volcanic aerosols (n=4), orbital parameters (n=3), solar irradiance (n=4), and changes in land surface properties resulting from land use (n=3)."*

L225: to improve clarity, extend this sentence with "...(>60%) ensemble members were not in drought at the same time as this would indicate….."
*Author response: We will add a new sentence at Line 225 stating "...in drought at the same time. Co-occurrence of droughts across different ensemble members would potentially indicate an externally forced component to drought occurrence."*

L233: replace "," with "…(101 years) differ and affect any disparity…."
*Author response: We will make the suggested edit.*

L235: for clarity, it would be worth reconfirming the number of distributions that are generated (i.e. 500)
*Author response: We will edit this sentence as follows: "...to create 500 distributions of possible values…"*

L255: do you mean "observed variability" instead of "MAP"?
*Author response: No - the CV metric compares interannual variability with MAP. So in this case, the models' simulated variability is too low compared with their simulated MAP. We will edit the text along the following lines to clarify this: "...i.e., the model interannual precipitation variability is too low compared with the model MAP…"*

L264: in addition to overall bias in mean MAP across the continent, the models also largely generate precipitation with reduced variability (with the exception of CSIRO-Mk31-1-2 and IPSL as previously stated).
*Author response: We will add the following qualifier at Line 264: "...despite overall bias in mean MAP across the continent, and generally too-low interannual variability."*

L280: to improve clarity, insert "across ensemble members" prior to referencing the supp figs.
*Author response: We will make the suggested edit.*

L285: does the statement "suggest similar spatial patterns" apply to all members, or just some? If this is across the ensemble, please state this up front in the paragraph as this would help demonstrate that the simulations are an adequate representation of the observations, which I believe is the intent of this paragraph, and it is key to providing a basis on which comparisons of HIST, piLM, and pi Control can be assessed. The message at present is a little
lost because the shortcoming are presented first and the purpose of the 20th century simulations is not clearly stated (i.e. for validating the model runs and providing credibility for the piLM and pi control runs).
*Author response: We will add the statement "In the multi-model mean, …" to the start of the paragraph (L277). We will also add that statement in Line 285/286.*

L311: particularly in southern and eastern Australia
*Author response: Thanks - we will make this correction.*

L323: for consistency, include "ranging from" or "range of" prior to "5.1-8"

*Author response: We will make the suggested edit.*

L325: Is this supposed to be "mean maximum drought length" for both metrics on this line?
*Author response: This should be 'maximum drought length', except where we are specifically referring to a multi-model mean. To make this clearer, we will change "length" in Line 324 to "lengths".*

L328: I'm not sure what is meant by "continent-spanning grids". Is this just "all locations"? Or "grids covering the mainland"?
*Author response: The former - the grids spanning all of Australia. Fair point - the 'Australian continent' technically includes New Guinea as well as mainland Australia and Tasmania. We will replace 'continent-spanning' with 'full' to differentiate this MDB-focussed analysis from the all-Australian-grid-cells analysis.*

L330: could you clarify which model simulations? I believe it would be the HIST model simulations
*Author response: Sorry for the confusion here. Yes, it is in the HIST simulations - this is stated at the start of the sentence "The relative intensity of 20th century droughts…". To improve clarity, we will add the word "HIST" before "model simulations" in Line 330.*

L332: add "%" to these numbers
*Author response: We will make this correction.*

L335-336: Does this statement not apply across the ensemble results too? Or is it just confined to the three best performing models?
*Author response: The difference is larger when only looking at the three 'best-performing' models, but yes - each model's HIST simulation generally shows more severe droughts than the same model's piLM simulations. We will remove the word "However" from Line 335.*

L336: "worse" is subjective. Use "more severe" or similar
*Author response: We will replace "worse" with "more intense/severe".*

L385: "The exception is volcanic forcing, where CESM LME most ensemble members in the CESM LME run with volcanic forcing are not in drought….". Also, it seems like a discussion of the agreement between ensemble members under LULC forcing is missing. It would also be good to comment on the variability of the forcing as it's very easy to see when volcanic forcing imposes a large change in the radiative forcing, but the variations in solar and LULC are less easy to identify.
*Author response: We see no major influence of LULC on multi-year drought occurrence in the MDB, so in the interests of brevity, do not discuss the forcings individually.*

Line 391: specify that this is in reference to results that are averaged across Australia.
*Author response: We will amend that sentence to state "*Overall our results suggest that across most of the continent, Australian droughts have not changed substantially in the last century compared to model simulations of the last millennium.*"

L415 to 417: Given the findings that 100-year samples result in different summary statistics compared to a single 1000 year record, these comparisons really should be made in the context of the distribution of 100 year samples taken from the longer record as opposed to comparing a 100 year long record with a 1000 year long record (i.e. fig 9a-d).
*Author response: In this situation, we may consider the 2d grids formed from the statistics of the 1000-year piLM simulations the 'control', and the 2d grids formed from the statistics of the 101-year HIST simulations the 'experiment'. The 1000-year piLM simulations show the climatological patterns we would expect today, without*

*the influence of anthropogenic forcing. We are assessing whether there is any* detectable *change, following the addition of anthropogenic forcings. If there is yet no detectable change, this could be 1) because there is no anthropogenic influence, or 2) because the anthropogenic influence is not yet large enough/has not had enough time to emerge from the range of internal variability. We state this at line 421: "*Hence, the lack of significant 20th century change across most of Australia in most drought metrics does not imply that there has been no human influence on Australian droughts during the 20th and 21st centuries, but rather that the 101-year HIST simulations are too short for significant differences to emerge.*"*

*To clarify, we will change '*significant differences*' to '*any significant anthropogenically-driven changes*' at line 424.*

L427: MDB (rather than MBD)
*Author response: Thank you for picking this up - we will make the suggested edit.*

L438: Can text be added to make this finding a bit more explicit? Such as: "The co-occurrence of volcanic eruptions and supressed drought conditions over the MDB appear to contradict existing understandings of the impacts of volcanic eruptions on El Niño-like conditions and subsequent impacts on rainfall in the MDB".
*Author response: Excellent suggestion. We will add the suggested sentence in place of the current sentence at Line 439-440. "*The co-occurrence of volcanic eruptions and suppressed drought conditions over the MDB appears to contradict current understanding of the impacts of eruptions on the El Niño-Southern Oscillation (ENSO), and subsequent impacts on rainfall in the MDB (Gillet et al., 2023)*"*

*Gillett, Z. E., Taschetto, A. S., Holgate, C. M., & Santoso, A. (2023). Linking ENSO to synoptic weather systems in eastern Australia. Geophysical Research Letters, 50, e2023GL104814. https://doi.org/10.1029/2023GL104814*

L 443-L445 needs to be clarified. At present it appears to contradict the first sentence of the conclusion.
*Author response: We will change "*exceptional*" in Line 444 to "*unprecedented*".*

---

## Author Comment (AC2)

**Author response:** We thank the Reviewer for their kind comments, and for their constructive review of our manuscript. We have addressed all suggestions (details below). In response to both these comments, and the suggestions of Reviewer 1, we will make the following changes—including clarifications to our methodology—to make our findings more robust:

- Add two new Supporting Figures:
  - One showing the return period of multi-year droughts in the Murray-Darling Basin, in each PMIP3 model's pre-industrial last millennium simulation
  - One showing the relative severity of the longest MDB drought in each simulation from each model, as well as observations
- Provide increased clarity around the interpretation of Supporting Figure 20, and also slightly modify this figure for ease of interpretation (details below)
- Provide more detail of calculation of the spatial correlations, and - at the Editor's discretion - either use these correlations to weight the calculation of multi-model means, or add a statement as to why we did not do this
- Add a statement as to why we did not bias-correct the models

Additionally, we will address all general and specific comments from the Reviewer as outlined below. We consider that these changes will result in a stronger paper with clearer, more robust findings.

**Review of 'Emerging anthropogenic influence on Australian multi-year droughts with potential for historically unprecedented megadroughts'**

The study focuses on deepening the understanding of the natural ranges of Australian droughts, which can be used to better assess intrinsic and externally forced drought risks in future planning. They address this by comparing droughts in the 20th century (1900-2000) based on observations and models, with simulated droughts during the pre-industrial millennium (850-1849). They seek to assess if drought characteristics (mean duration, maximum length, intensity, etc.) have changed during the last century, compared to the past millennium.

One of the main conclusions is that multi-year droughts have been longer on average during the 20th century over part of Australia (including the MDB), compared to the last millennium, and that anthropogenic forcing is the likely cause of this change.

The authors also conclude that having a larger sample (pre-industrial millennium) allows for a better characterization of natural drought variability, reaching out extreme events (longest droughts) that have not been observed over the last century. Based on this, the authors conclude that such extreme events are part of the natural range of droughts in Australia, and thus they can be expected in the future. This, superimposed with projected drying trends, pose critical challenges for adaptation planning.

The article is well written and the motivation and research question is clear. However, there are some methodological aspects that should be addressed before drawing robust conclusions:

1) There are models that perform better than others during the historical period, and this is quantified as part of the analysis (Sect. 2.2.1; 2.3.3; Supporting Fig. 19). In this line, the interpretation of results should also account for these different performances. We should trust more those models that better represent the observations in the historical period, right?

For example, if the physical mechanisms represented by a particular model structure leads to lower interannual precipitation variability compared to observations, it is expected that its simulations during the last millennium

reflect the same bias, and vice versa for the case of higher interannual variability. However, the conclusions of the paper are based on the average of all models, independently of their performances during the historical period.

*Author response: We did not weight the multi-model averages as there is evidence from studies analysing future projections that weighting for model performance does not necessarily produce better projections (e.g., Abramowitz et al., 2019). Such weighting approaches can be problematic as they can lead to results that are heavily weighted towards a small number of highly dependent models and there is no universally agreed way to do such weighting (Eyring et al., 2019). There are additional considerations in our case, including:*

1. *there are many non-significant correlations, meaning the values are not necessarily meaningful, and*
2. *as we state in the paper (and show in Supp. Fig. 20), in the short 101-year time period of comparison, there is a high random element to the multi-year drought metrics.*

*This means that the weighting may not necessarily be a fair representation of each model's long-term skill in simulating multi-year drought characteristics.*

*Given this context, **at the Editor's discretion** we will re-calculate the ensemble-mean values (shown in Figs. 5-7), weighting according to each model's correlation with observations of that same metric. If not, we will add a statement to the end of Section 2.3.2 stating:*

*"We calculated arithmetic multi-model means rather than weighting the models according to the spatial correlations. Evidence from future projections suggests that weighting for model performance does not necessarily produce better projections (e.g., Abramowitz et al., 2019). Additionally, weighting can lead to results that are heavily weighted towards a small number of highly dependent models and there is no universally agreed way to do such weighting (Eyring et al., 2019)."*

*Abramowitz, G., Herger, N., Gutmann, E., Hammerling, D., Knutti, R., Leduc, M., Lorenz, R., Pincus, R., and Schmidt, G. A.: ESD Reviews: Model dependence in multi-model climate ensembles: weighting, sub-selection and out-of-sample testing, Earth Syst. Dynam., 10, 91–105, https://doi.org/10.5194/esd-10-91-2019, 2019.*

*Eyring, V., Cox, P.M., Flato, G.M. et al. Taking climate model evaluation to the next level. Nature Clim Change 9, 102–110 (2019). https://doi.org/10.1038/s41558-018-0355-y*

Given the large differences between models (spatial precipitation patterns, MAP values and performance against observations), I don't think mean ensemble values can be directly used for interpretation of results. For example, the assessment of the "Possible anthropogenic influence on Australian multi-year droughts" (Section 3.4) relies on the ensemble mean of models, however, we know that there are models that perform better than others.

*Author response: Analysis of ensemble mean values is extremely common, particularly when using CMIP/PMIP ensembles; it is a straightforward way of identifying common (and therefore climatically meaningful) signal whilst smoothing over individual model biases. Additionally, we do not purely rely on the multi-model mean, but also show all individual model results in the Supplementary Information. If we weight the means according to the Reviewer's request, the ensemble-mean values will more closely reflect values of the models with better apparent skill in this historical period. However, this is not necessarily a guarantee of more accurate results, for the reasons outlined above.*

A way to account for these different model performances could be to apply a statistical correction to the models before analyzing droughts, similarly than those applied to GCMs in the historical period before analyzing their future projections (e.g., Cannon, 2018 and references therein). This data-process involves that each model is

corrected according to their own performances in the historical period, and then results can be interpreted similarly across models.

*Author response: There are multiple different ways to bias-correct and the choice of correction method is subjective and can strongly influence the results. Using CMIP5 simulations over Australia, Vogel et al. (2023) for example recently showed that different bias correction methods can lead to large differences in simulated rainfall climatology, variability and extremes. Furthermore, no single bias correction method was able to outperform the others when evaluated for multiple metrics. Choosing one bias correction method for our study would thus be highly subjective and a comparison of multiple methods is out of the scope of this study. We also note that our method accounts for some of the differences in the simulated MAP by applying the drought metrics separately to each model's own climatology (such that all drought metrics are calculated relative to the model's own mean).*

*To clarify this, we will add the following statement at Line 150: "Note that we do not use the results of this verification to bias-correct the models. Vogel et al. (2023) used CMIP5 simulations of Australian precipitation to demonstrate that different bias correction methods can lead to large differences in simulated rainfall climatology, variability and extremes, and that no single bias correction method outperforms the others when evaluated for multiple metrics."*

*Vogel, E., F. Johnson, L. Marshall, U. Bende-Michl, L. Wilson, J. R. Peter, C. Wasko, S. Srikanthan, W. Sharples, A. Dowdy, P. Hope, Z. Khan, R. Mehrotra, A. Sharma, V. Matic, A. Oke, M. Turner, S. Thomas, C. Donnelly, V. C. Duong, An evaluation framework for downscaling and bias correction in climate change impact studies, Journal of Hydrology, Volume 622, Part A, 2023. https://doi.org/10.1016/j.jhydrol.2023.129693*

From Sect. 2.3.2, it is inferred that droughts are defined as deviations from the climatology of each model (right?) If the models are bias corrected, the same climatological mean (that from AWAP) could be used for drought definition. And direct comparison between models could be applied, instead of %. This is easier for interpretation than "For example, 0% represents the climatological mean precipitation, and 100% represents zero precipitation". Same for severity, it would be much easier to compare directly mm across models, instead of % ("For example, a value of 200% represents a total deficit equal to two years of mean precipitation.")

Comparing deviation metrics as % is influenced by the native MAP of each model (deviations from a low absolute MAP values represent larger % than when the MAP is larger). By comparing Fig 2.a and Fig. 3, it can be seen that some models have MAP biases up to 100%, with similar absolute biases that observed MAP in Fig. 2a.

*Author response: In the case of our analysis, all drought metrics are calculated relative to the models' own climatologies. Therefore, correcting for MAP biases would have a minimal impact on the results as most of the drought metrics presented are based on years above/below the models' own climatological means. The Reviewer is correct that bias-correcting the mean would allow a direct comparison in mm but the relative intensity metric used here in fact already allows for direct comparison across models. Changing this to mm would not change the relative differences across models, and hence would not affect our results or conclusions.*

*Cannon, A.J. Multivariate quantile mapping bias correction: an N-dimensional probability density function transform for climate model simulations of multiple variables. Clim Dyn 50, 31–49 (2018). https://doi.org/10.1007/s00382-017-3580-6*

2) Having a more extreme event in a large sample can be somehow expected, but I am missing an assessment of the return period of such events. The longest droughts simulated over the pre-industrial millennium, can be expected to happen over the next century, couple of centuries, thousand years?

*Author response: We will include the following plot as an additional Supporting Figure, showing the return period of multi-year droughts in the Murray-Darling Basin, in each of the PMIP3 models' piLM simulations. In most models, the longest drought occurs only once, giving an estimated return period of around 1000 years (the length of the simulations). However, most models still simulate very long MDB droughts (10-20 years) with relatively short return periods of ~100-150 years. We will add a brief discussion of these results in Section 3.3*

[Figure]

In the same line, I think that for providing evidence for adaptation planning, the longest droughts should be assessed in conjunction with their deficits: it is not the same to communicate that 20-years of minor droughts (e.g., 0-10% deficits) can be expected that to communicate that 20-years of severe droughts (e.g., >40% deficits) can be expected in the future. This could be done by accounting for relative severity together with maximum length.

*Author response: Given this adaptation-focussed analysis is most likely to be useful for the Murray-Darling Basin, we will add a new Supporting Figure showing the relative severity associated with the longest drought. The figure will be similar to what is shown on the following page, where tile colour corresponds to the relative severity of the longest drought, and the text annotation states the length of that drought, in years. As well as the necessary additions to the Methods and Results, we will state in the Discussion that although the piLM and piControl simulations generally produce longer maximum MDB drought length than the HIST simulations, they are not always more severe in terms of total deficits throughout the drought (in the case of this simple comparison).*

[Figure]

**Minors comments:**

Supp. Fig. 7: "Mean multi-year drought length in (a) observations (1900-2000) and (b-l) model simulations of the pre-industrial last millennium (850-1849). Showing the CESM LME ensemble mean." It should say, panel Fig. 7l presents the CESM LME ensemble mean. Same for all figures.

*Author response: Thank you for catching this. We will modify the captions for Supporting Figs 1–18 to clarify this.*

Title: it is a complicated title that I don't think is communicating the main messages of the paper. I recommend the authors to consider a simpler one.

*Author response: We agree that it is quite a long title(!) however we consider that it does convey our main takeaway messages. An alternate version could be "Australian multi-year droughts show emerging anthropogenic change with potential for megadroughts that exceed recent historical experience", however that is not much less complicated. Alternatively "Australian multi-year droughts show high variability across centuries and an emerging anthropogenic influence", however we would like to emphasise the (novel, policy-relevant) finding that natural variability in Australian droughts can produce droughts that exceed historical experience. Of these two alternate titles, the first is our preference. We would be more than happy to work with the Editor in coming up with a shorter title.*

Abstract: "Model simulations suggest future droughts across Australia could be much longer than what has been experienced in the twentieth century, even without any human influence." This can be misunderstood as future projections, please re-phrase. An option could be: Drought simulations over the last millennium suggests that future droughts across Australia could be much longer than what has been experienced in the twentieth century, even without any human influence.

*Author response: We will modify the sentence to "Model simulations of droughts over the past millennium suggest future droughts across Australia could be much longer than what has been experienced in the twentieth century, even without any human influence".*

---

## Author Response (AR1)

Response to Editor and Referees for the manuscript "**Emerging anthropogenic influence on Australian multi-year droughts with potential for historically unprecedented megadroughts**", submitted to *Hydrology and Earth System Sciences* by Falster and Wright et al.

Throughout, comments from the Editor and Reviewers are in black, and author responses are in red.
* * *
Dear Authors,

The two referees have expressed appreciation for your work, that is indeed extremely interesting given the importance of better understanding the natural variability of droughts and megadroughts and the potential for the increase of their severity due to anthropogenic impacts (particularly significant for Australia, but applicable in other regions).

The referees, that I warmly thank, have timely contributed to the open discussion with very detailed and constructive comments on your work, and from your replies I am sure that following their suggestions, your analysis will further improve, clarifying the methodology (including more detail on the spatial correlation analysis) and providing further analyses and figures including the most recent droughts and additional interpretation of the results (see information on return periods and on relative severity).

I therefore invite you to proceed with the revision, following the useful advice provided by the Referees, hoping that they may find the time to give me their valuable opinions also on the new version of the manuscript.

Concerning the issue of averaging of the model outputs: even I fully agree with Ref2 that not all climatological models provide reliable estimates on the control period (the way forward would be to improve the models…), I concur with the Authors that in the climate modelling community there is more consensus on not weighting the outputs. I also agree that in this kind of analysis, where the series are not directly used in subsequent impact studies (such as use as input into hydrological models etc), bias-correction procedures are not strictly necessary, I believe that working on the "delta", i.e. the differences between the control/observation and other periods, already allows a direct comparison across models.

Concerning the title: I agree with Ref#2 that the current title may be more effective for communicating the focus of your work, and I add another possible alternative to the Authors' proposals for changing it: "Combination of natural variability and emerging anthropogenic influence shows potential for historically unprecedented megadroughts in Australia".

Best wishes,
Elena Toth
HESS Editor

**Author response:** We thank the Editor and reviewers for their kind comments, and for their valuable and highly constructive reviews of our manuscript. We have addressed all suggestions from both reviewers and the Editor (details below). In response to these comments, we have made the following changes to improve the clarity of our manuscript:

- Provided an alternative title for the paper
- Add two new Supporting Figures:
  - One showing the return period of multi-year droughts in the Murray-Darling Basin, in each PMIP3 model's pre-industrial last millennium simulation
  - One showing the relative severity of the longest droughts in each simulation from each model, as well as observations

- Provide increased clarity around the interpretation of Supporting Figure 20 (now Supp. Fig. 22), and also slightly modify this figure for ease of interpretation (details below)
- Provided more detail of calculation of the spatial correlations and added a statement as to why we did not use these correlations to weight the calculation of multi-model means
- Added a statement as to why we did not bias-correct the models

We also provided two new analyses that are publicly available in our responses to the Reviewers:
- A comparison of multi-year drought characteristics in the observational dataset used in this paper with observations that extend to 2021, thereby encompassing the Millennium and Tinderbox droughts.
- For each model, a timeseries of the *maximum* relative intensity of each drought across 850-2000 CE for the MDB.

Aside from making changes suggested by the Reviewers, we have made three minor corrections:
- In all cases, changed 'CESM LME' to 'CESM-LME'.
- Removed the line '*Showing the CESM-LME ensemble mean*' from Supporting Figures 13-18 (as this is a control run).
- Added a brief extra statement at Line 543, about palaeoclimate context provided by the Law Dome ice core sea salt record

Below, we state how we have addressed all general and specific comments from the Reviewers. We consider that the changes suggested by the Reviewers result in a stronger paper with clearer, more robust findings. For ease of comparison, references to line numbers correspond to the 'tracked change' manuscript.

**Reviewer 1**

**Review of 'Emerging anthropogenic influence on Australian multi-year droughts with potential for historically unprecedented megadroughts'**

The authors have presented a well-motivated and, largely, clearly executed analysis of changes in large droughts in Australia using validated modelled drought outputs for the historical period and over the last millennium as derived from paleoclimate data. I believe that this manuscript is a worthy contribution to the scientific discourse, but I believe some revisions and additional analysis is required to make this a robust study as follows:

- My most pressing concern is ensuring that the validation is sound and appropriately quantified;
- At present, I do not believe that the presentation of figures intended for communicating the impact of different sample sizes is sufficient; and
- I feel that a measure of drought intensity that is comparable across events of different length is missing
- More detail for these three dot points are provided under the general comments.
- And finally, I would like to see an acknowledgement of the limitations in making comparisons with historical droughts, particularly since the analysis omits most of the millennium and all of the Tinderbox droughts.

*Author response: Regarding the last dot point, which is not addressed under 'General comments': we re-calculated all multi-year drought metrics, but using a version of the observational dataset that extends to the year 2021 (thereby including the Millennium and Tinderbox droughts). We provide that comparison on the following page. For mean and maximum drought length, relative severity and intensity, and proportion of time spent in drought, differences between the two datasets are negligible. Drought frequency increases slightly with the addition of the extra 21 years - particularly in eastern Australia.*

*We added the following statement at Line 513: "We note that the PMIP3 past1000 simulations do not cover two of the MDB's most impactful droughts of the historical period: the Millennium and 2017-2019 Tinderbox droughts.*

However, the occurrence of two major droughts in the first two decades of the 21st century provides additional support for our finding that the MDB is spending more time in drought during the historical period compared with natural variability during the pre-industrial last millennium."

*We also added the following statement at Line 496: "*However, calculating observed multi-year drought metrics on observations extending to 2021 (i.e., encompassing the Millennium and Tinderbox droughts) results in negligible changes to mean and maximum drought length, relative severity and intensity, and proportion of time spent in drought (not shown). Drought frequency increases slightly with the addition of the extra 21 years—particularly in eastern Australia (not shown)."

[Figure]

[Figure]

Otherwise, there are a small number of clarifications, particularly of the caveats, and these are detailed in the minor comments.

**General comments**
A brief discussion is needed to acknowledge the limitation of using one definition of a water year for all of Australia and locations where the water year definition used is most/least relevant.

*Author response: We have added the following at Line 110: "*Precipitation seasonality varies across Australia. For example, in tropical northern Australia, 'years' may be better represented by 'tropical years', where each year starts, for example, in May and finishes in April of the following year. However, this is not applicable across the entire continent, where some regions have winter-dominated precipitation, and others have no distinct seasonality. Given this study focuses on multi-year events, our choice of a calendar year for calculation of annual totals should not have a major influence on the results."

I need to see more detail about how the spatial correlations were calculated. Was this a calculation of correlations between matching locations that were then averaged across the region, or was the significance at each location assessed independently, or was a field significance considered and if so, which method was used (e.g. false discovery rate, walker's test, counting test)? The latter (a measure of field significance) is what is required. An averaging of correlation results is not appropriate, and assuming spatial independence is also inappropriate. The results of quantifying the field significance of the similarities between the observed and modelled droughts will impact on the credibility and interpretability of the remaining results. If fiend significance results markedly alter the validity of the modelled results, the interpretation of the pre-industrial millennia results will need to be re-interpreted accordingly.

*Author response: For Supp. Figs. 1-6, we show a pattern correlation. We have added the following, more detailed, description to the Methods (at Line 162):* "Spatial correlations between pairs of two-dimensional grids were calculated by flattening each grid, resulting in two directly comparable vectors, with each index position of each vector representing the values at a particular latitude-longitude pair. We calculated the Pearson correlation coefficient, and provide an estimate of the significance of that coefficient (reported 'significant' if p < 0.05)."

*For Fig. 9 where we calculate significance per pixel, we have now adjusted the p-values as suggested (described at L230 and in the figure caption).*

The measures of both relative drought intensity and severity appear to be functions of the average deviation from climatology across the event. It appears to me that relative drought severity is a superfluous metric since drought length is also presented (although the figures show the mean over time and sometimes across models, so I realise it is showing something different then taking the product of, say, fig 5b and fig 6b).

*Author response: That is correct - the 'drought severity' metric combines the information from the drought length and drought intensity metrics; drought intensity is independent from duration. The severity metric provides an estimate of the* total *precipitation deficit over the drought, which is not easily apparent from inspecting the length and intensity metrics separately. We consider this to be an important aspect of the drought characterisation processes, as it provides insight into the total water stress placed on the region over the span of a particular drought. We also chose this particular metric of drought severity as it is consistent with our other metrics (e.g. drought mean and maximum length), which are expressed for the whole event.*

What seems to be missing is a measure that reflects the most severe drought year (or years) such as the most intense two consecutive years of drought within an event and to see if the maximum (annual or consecutive multi-annual) intensity is changing in different time periods. A measure like this would prevent the metric from

being influenced by the definition of the event duration, which is the case for drought intensity and severity metrics, particularly since event length would be sensitive to the definition of drought in determining onset and termination.

*Drought severity is indeed dependent on the metric used to identify droughts. In the '2S2E' method used in this paper, multi-year droughts would be longer and more severe (but less intense) compared with, for example, a method where 'multi-year droughts' are two or more years below the 20th percentile of climatological precipitation. Given the already very large number of figures in this paper, we do not provide a test of the sensitivity of our results to different drought identification methods. However, we have corrected 'intensity' at Line 180 to 'severity' (as assessed by Cook et al., 2022).*

*Whilst we agree that the new analysis proposed by the Reviewer is certainly an interesting one, assessing temporal variability in maximum drought intensity is not practical for a continent-wide analysis. Here we have performed this analysis for the Murray-Darling Basin: the below figure shows, for each model, the maximum relative intensity of each drought, for 850-2000 - i.e. the single most intense year within the drought (as proposed by the Reviewer). Across the PMIP3 models, there is no consistent temporal variability in drought maximum intensity through time.*

[Figure]

Section 3.2.3. Results of Fig 6 b-d and f-h are presented in the text, but references to the figures need to be made. I recognise that reference is made to supp. Figs 1-2 and 5-6 that reflect the points made in the text in more detail, but Fig 6 b-d and f-h are still relevant and need to be referenced in the text.

*Author response: We have made the following changes to ensure specific reference to Fig 6b-d and f-h:*
- *At Line 328, we reference Fig. 6b and f at the end of the line.*
- *At Line 332, we have edited the figure reference to state 'Figs. 5-7a,e compared with Figs 5-7c,d,g,h'. We also added the word 'broadly' before 'resemble' in Line 331*

The specification of significance level needs to be stated in terms of what significance level has been chosen for the evaluation i.e. = 0.01, rather than reporting overall p-values.

*Author response: We now state at L166 that we used α = 0.05 as our significance threshold.*

Section 3.3. I would like some text around how the precipitation mean and variance compare between modelled and observed specifically in MDB (rather than relying on the reader to interpret the figures themselves) as this could help explain the difference in drought results.

*Author response: We added this new sub-section at Line 302, and hence removed the previous sentence about CV in the MDB at Line L290*

Section 3.3.2. and supporting fig 20. It does not follow that large spatial variability implies anything about the adequacy of record length. Another justification is needed here.

*Author response: Major and random spatial variability, rather than well-defined spatial patterns, implies a large random element to the patterns. In this particular context, this suggests that there are insufficient multi-year droughts for a climatological pattern to emerge. This points to an important role of model internal variability at these time scales as the different ensemble members discussed here only differ in their internal variability (rather than other factors such as model physics). To clarify, we have added the following text at Line 409:* "That is, the presence of random spatial variability, rather than well-defined spatial patterns, implies a large random element. In this context, this suggests that there are insufficient multi-year droughts in a 101-year sample for a climatological pattern to emerge."

Also, I can see the intent of what the authors are aiming to communicate here: that shorter simulations fail to fully explore how variable drought can be given the range of drought conditions that can be explained over a longer period. I think Supp Fig 20 b is sufficient because it is clear that the maximum drought length obtained from a shorter 101-year sample will likely underestimate the maximum plausible drought length. However, I don't think the remaining plots in this figure demonstrate that the drought characteristics sampled in a 101 year-long sample are not representative of what could be reasonably expected in our climate and an alternative way of presenting this data is needed.

*Author response: Supporting Figure 20 (now Supp. Fig. 22) demonstrates that no single 101-year segment can capture the full range of variability present in a 1000-year sample. Hence, any single 101-year segment (such as the historical period) likely gives a skewed representation of long-term variability in that particular drought metric. This is true for all metrics, not just maximum possible drought length.*

*We recognise that by showing the overall mean values from the piLM simulations as blue dots on Supp. Fig. 22, we did not illustrate this point particularly well. We have now added dots showing the equivalent values from each model's* **HIST** *simulation. We also added the following text to Section 3.3.2:* "That is, individual 101-year segments do not capture the full range of variability represented by the 1000-year piLM period, with the magnitude of drought durations, intensity, severity, frequency, and proportion of time spent in drought varying markedly from one 101-year period to another. This means that selecting any single 101-year period (such as the historical period) is not representative of the full variability in the models' simulated precipitation (blue dots in Supp. Fig. 22)."

One suggestion I have would be to plot a cumulative density line or scatter for each of the 500 samples on a single plot and then overlayed would be the cumulative density line for the 1000-year long simulation. As a dummy example, I've done this for 500 samples of n=100 for a normal distribution of mean=10 and sd=2 (these values have no meaning or significance, it's just for an example) with an additional sample of n=1000 shown in bold. The historical or HIST ecdf could also be added and discussed with respect to over/underestimating flood characteristics at different magnitudes with respect to the longer record. I'd be more than happy for the authors to either adopt this or develop an alternative for presenting their findings that would provide a figure that supports the argument they are making in the text of section 3.3.2.

*Author response: Thank you for the excellent suggestion. In response to this comment, as well as a comment from Reviewer 2, we have now added a new Supporting Figure showing the return period of droughts of different lengths in the MDB, from each model's piLM run (Supp. Fig. 21). However, showing return periods calculated from the 101-year segments on that same plot is slightly misleading. For example, the longest single drought of the piLM simulation will occur only once in the piLM run, giving it a return period of ~1000 years, while in the 101-year segment containing that same drought, it will have a return period of ~101 years. This makes it hard to*

*compare the two analyses on the sample plot and may be misinterpreted by readers. This type of analysis is also slightly complicated by the fact that multi-year droughts are discrete events that do not occur every year.*

*In showing values for the HIST simulations on Supp. Fig. 22 (as detailed above), we consider that we more clearly demonstrate that using a too-short time period leads to an inaccurate understanding of multi-year drought characteristics. Combined with our additional text as shown above, this better supports our arguments in section 3.3.2.*

**Minor comments:**

L48: A reference is needed for the Tinderbox drought being a "major" drought.

*Author response: We have added the following reference: Nguyen, H., M. C. Wheeler, H. H. Hendon, E.-P. Lim, J. A. Otkin, The 2019 flash droughts in subtropical eastern Australia and their association with large-scale climate drivers, Weather and Climate Extremes, Volume 32, 2021. https://doi.org/10.1016/j.wace.2021.100321.*

L102: Full stop at the end of this sentence

*Author response: Thank you for catching this. We have added this full stop.*

L105: Use "0.05° × 0.05° latitude/longitude resolution" for consistency with later model resolution descriptions.

*Author response: We have made the suggested edit.*

L146: clarify that the bias relative to observations is shown for each member as well as the overall ensemble mean.

*Author response: We have edited the sentence to "For each model, we show the results as absolute bias relative to observations."*

L193 and elsewhere: specify that the resolution is for latitude/longitude

*Author response: We have edited this and similar sentences to "...into 2° x 2° (latitude x longitude) resolution…" - e.g. L212.*

L209: is the percentage bias also reported for the rest of Australia? If not, why not?

*Author response: We only calculate the percent bias values for the Murray-Darling Basin. This is because the MDB analyses are performed on a single area-mean precipitation timeseries (rather than each individual grid cell), which permits these more detailed analyses. We feel this is less appropriate if calculated on area-mean precipitation over the whole continent due to the many climate zones this would encompass.*

L222: could you clarify in L126 how many members are run in natural or fully forced or single forcing so it is clear what this "30" is based on?

*Author response: We have added this information at Lines 138 in the following format:* "...well-mixed greenhouse gases (n=3), volcanic aerosols (n=4), orbital parameters (n=3), solar irradiance (n=4), and changes in land surface properties resulting from land use (n=3)."

L225: to improve clarity, extend this sentence with "...(>60%) ensemble members were not in drought at the same time as this would indicate….."

*Author response: We have added a new sentence at Line 254 stating "...in drought at the same time. Co-occurrence of droughts across different ensemble members would potentially indicate an externally forced component to drought occurrence."*

L233: replace "," with "…(101 years) differ and affect any disparity…."

*Author response: We have made the suggested edit.*

L235: for clarity, it would be worth reconfirming the number of distributions that are generated (i.e. 500)
*Author response: We have edited this sentence as follows: "...to create 500 distributions of possible values…"*

L255: do you mean "observed variability" instead of "MAP"?
*Author response: No - the CV metric compares interannual variability with MAP. So in this case, the models' simulated variability is too low compared with their simulated MAP. We have therefore edited the text at L291 to clarify this: "...i.e., the model interannual precipitation variability is too low compared with the model MAP…"*

L264: in addition to overall bias in mean MAP across the continent, the models also largely generate precipitation with reduced variability (with the exception of CSIRO-Mk31-1-2 and IPSL as previously stated).
*Author response: We have added the following qualifier at Line 300: "...despite overall bias in mean MAP across the continent, and generally too-low interannual variability."*

L280: to improve clarity, insert "across ensemble members" prior to referencing the supp figs.
*Author response: We have made the suggested edit.*

L285: does the statement "suggest similar spatial patterns" apply to all members, or just some? If this is across the ensemble, please state this up front in the paragraph as this would help demonstrate that the simulations are an adequate representation of the observations, which I believe is the intent of this paragraph, and it is key to providing a basis on which comparisons of HIST, piLM, and pi Control can be assessed. The message at present is a little lost because the shortcoming are presented first and the purpose of the 20th century simulations is not clearly stated (i.e. for validating the model runs and providing credibility for the piLM and pi control runs).
*Author response: We have added the statement "In the multi-model mean, …" to the start of the paragraph (L321). We also added that statement in Line 327.*

L311: particularly in southern and eastern Australia
*Author response: Thanks - we have made this correction.*

L323: for consistency, include "ranging from" or "range of" prior to "5.1-8"
*Author response: We have made the suggested edit.*

L325: Is this supposed to be "mean maximum drought length" for both metrics on this line?
*Author response: This should be 'maximum drought length', except where we are specifically referring to a multi-model mean. To make this clearer, we have changed "length" in Line 368 to "lengths".*

L328: I'm not sure what is meant by "continent-spanning grids". Is this just "all locations"? Or "grids covering the mainland"?
*Author response: The former - the grids spanning all of Australia. Fair point - the 'Australian continent' technically includes New Guinea as well as mainland Australia and Tasmania. We have replaced 'continent-spanning' with 'full' to differentiate this MDB-focussed analysis from the all-Australian-grid-cells analysis.*

L330: could you clarify which model simulations? I believe it would be the HIST model simulations
*Author response: Sorry for the confusion here. Yes, it is in the HIST simulations - this is stated at the start of the sentence "The relative intensity of 20th century droughts…". To improve clarity, we have added the word "HIST" before "model simulations" in Line 374.*

L332: add "%" to these numbers
*Author response: We have made this correction.*

L335-336: Does this statement not apply across the ensemble results too? Or is it just confined to the three best performing models?

*Author response: The difference is larger when only looking at the three 'best-performing' models, but yes - each model's HIST simulation generally shows more severe droughts than the same model's piLM simulations. We have removed the word "However" from Line 379.*

L336: "worse" is subjective. Use "more severe" or similar

*Author response: We have replaced "worse" with "more intense/severe".*

L385: "The exception is volcanic forcing, where CESM LME most ensemble members in the CESM LME run with volcanic forcing are not in drought….". Also, it seems like a discussion of the agreement between ensemble members under LULC forcing is missing. It would also be good to comment on the variability of the forcing as it's very easy to see when volcanic forcing imposes a large change in the radiative forcing, but the variations in solar and LULC are less easy to identify.

*Author response: We see no major influence of LULC on multi-year drought occurrence in the MDB, so in the interests of brevity, do not discuss the forcings individually.*

Line 391: specify that this is in reference to results that are averaged across Australia.

*Author response: We have amended that sentence (L459) to state "Overall, our results suggest that across most of the continent, Australian droughts have not changed substantially in the last century compared to model simulations of the last millennium."*

L415 to 417: Given the findings that 100-year samples result in different summary statistics compared to a single 1000 year record, these comparisons really should be made in the context of the distribution of 100 year samples taken from the longer record as opposed to comparing a 100 year long record with a 1000 year long record (i.e. fig 9a-d).

*Author response: In this situation, we may consider the 2d grids formed from the statistics of the 1000-year piLM simulations the 'control', and the 2d grids formed from the statistics of the 101-year HIST simulations the 'experiment'. The 1000-year piLM simulations show the climatological patterns we would expect today, without the influence of anthropogenic forcing. We are assessing whether there is any detectable change, following the addition of anthropogenic forcings. If there is yet no detectable change, this could be 1) because there is no anthropogenic influence, or 2) because the anthropogenic influence is not yet large enough/has not had enough time to emerge from the range of internal variability. We state this at line 491: "Hence, the lack of significant 20th century change across most of Australia in most drought metrics does not imply that there has been no human influence on Australian droughts during the 20th and 21st centuries, but rather that the 101-year HIST simulations are too short for significant differences to emerge."*

*To clarify, we have changed 'significant differences' to 'any significant anthropogenically-driven changes' at line 494.*

L427: MDB (rather than MBD)

*Author response: Thank you for picking this up - we have made the suggested edit.*

L438: Can text be added to make this finding a bit more explicit? Such as: "The co-occurrence of volcanic eruptions and supressed drought conditions over the MDB appear to contradict existing understandings of the impacts of volcanic eruptions on El Niño-like conditions and subsequent impacts on rainfall in the MDB".

*Author response: We have added the following sentence in place of the previous sentence (Line 526): "The co-occurrence of volcanic eruptions and suppressed drought conditions over the MDB appears to contradict current*

understanding of the impacts of eruptions on the El Niño-Southern Oscillation (ENSO), and subsequent impacts on rainfall in the MDB (Gillet et al., 2023)."

*Gillett, Z. E., Taschetto, A. S., Holgate, C. M., & Santoso, A. (2023). Linking ENSO to synoptic weather systems in eastern Australia. Geophysical Research Letters, 50, e2023GL104814. https://doi.org/10.1029/2023GL104814*

L 443-L445 needs to be clarified. At present it appears to contradict the first sentence of the conclusion.
*Author response: We have changed "exceptional" in Line 535 to "unprecedented".*

**Reviewer 2**
**Review of 'Emerging anthropogenic influence on Australian multi-year droughts with potential for historically unprecedented megadroughts'**
The study focuses on deepening the understanding of the natural ranges of Australian droughts, which can be used to better assess intrinsic and externally forced drought risks in future planning. They address this by comparing droughts in the 20th century (1900-2000) based on observations and models, with simulated droughts during the pre-industrial millennium (850-1849). They seek to assess if drought characteristics (mean duration, maximum length, intensity, etc.) have changed during the last century, compared to the past millennium.

One of the main conclusions is that multi-year droughts have been longer on average during the 20th century over part of Australia (including the MDB), compared to the last millennium, and that anthropogenic forcing is the likely cause of this change.

The authors also conclude that having a larger sample (pre-industrial millennium) allows for a better characterization of natural drought variability, reaching out extreme events (longest droughts) that have not been observed over the last century. Based on this, the authors conclude that such extreme events are part of the natural range of droughts in Australia, and thus they can be expected in the future. This, superimposed with projected drying trends, pose critical challenges for adaptation planning.

The article is well written and the motivation and research question is clear. However, there are some methodological aspects that should be addressed before drawing robust conclusions:

> 1) There are models that perform better than others during the historical period, and this is quantified as part of the analysis (Sect. 2.2.1; 2.3.3; Supporting Fig. 19). In this line, the interpretation of results should also account for these different performances. We should trust more those models that better represent the observations in the historical period, right?

> For example, if the physical mechanisms represented by a particular model structure leads to lower interannual precipitation variability compared to observations, it is expected that its simulations during the last millennium reflect the same bias, and vice versa for the case of higher interannual variability. However, the conclusions of the paper are based on the average of all models, independently of their performances during the historical period.

*Author response: We did not weight the multi-model averages as there is evidence from studies analysing future projections that weighting for model performance does not necessarily produce better projections (e.g., Abramowitz et al., 2019). Such weighting approaches can be problematic as they can lead to results that are heavily weighted towards a small number of highly dependent models and there is no universally agreed way to do such weighting (Eyring et al., 2019). There are additional considerations in our case, including:*
> *1. there are many non-significant correlations, meaning the values are not necessarily meaningful, and*

2.  *as we state in the paper (and show in Supp. Fig. 20), in the short 101-year time period of comparison,
    there is a high random element to the multi-year drought metrics.*

*This means that the weighting may not necessarily be a fair representation of each model's long-term skill in
simulating multi-year drought characteristics.*

*Accordingly, we have added a statement to the end of Section 2.3.2 stating:* "We calculated arithmetic
multi-model means rather than weighting the models according to the spatial correlations. Evidence from future
projections suggests that weighting for model performance does not necessarily produce better projections (e.g.,
Abramowitz et al., 2019). Additionally, weighting can lead to results that are heavily weighted towards a small
number of highly dependent models and there is no universally agreed way to do such weighting (Eyring et al.,
2019)."

*Abramowitz, G., Herger, N., Gutmann, E., Hammerling, D., Knutti, R., Leduc, M., Lorenz, R., Pincus, R., and
Schmidt, G. A.: ESD Reviews: Model dependence in multi-model climate ensembles: weighting, sub-selection
and out-of-sample testing, Earth Syst. Dynam., 10, 91–105, https://doi.org/10.5194/esd-10-91-2019, 2019.*

*Eyring, V., Cox, P.M., Flato, G.M. et al. Taking climate model evaluation to the next level. Nature Clim Change
9, 102–110 (2019). https://doi.org/10.1038/s41558-018-0355-y*

Given the large differences between models (spatial precipitation patterns, MAP values and performance
against observations), I don't think mean ensemble values can be directly used for interpretation of
results. For example, the assessment of the "Possible anthropogenic influence on Australian multi-year
droughts" (Section 3.4) relies on the ensemble mean of models, however, we know that there are models
that perform better than others.

*Author response: Analysis of ensemble mean values is common, particularly when using CMIP/PMIP
ensembles; it is a straightforward way of identifying common (and therefore climatically meaningful) signal
whilst smoothing over individual model biases. Additionally, we do not purely rely on the multi-model mean, but
also show all individual model results in the Supplementary Information. As stated by the Editor, the way
forward here, ultimately, is to improve the models(!)*

A way to account for these different model performances could be to apply a statistical correction to the
models before analyzing droughts, similarly than those applied to GCMs in the historical period before
analyzing their future projections (e.g., Cannon, 2018 and references therein). This data-process involves
that each model is corrected according to their own performances in the historical period, and then results
can be interpreted similarly across models.

*Author response: There are multiple different ways to bias-correct and the choice of correction method is
subjective and can strongly influence the results. Using CMIP5 simulations over Australia, Vogel et al. (2023)
for example recently showed that different bias correction methods can lead to large differences in simulated
rainfall climatology, variability and extremes. Furthermore, no single bias correction method was able to
outperform the others when evaluated for multiple metrics. Choosing one bias correction method for our study
would thus be highly subjective and a comparison of multiple methods is out of the scope of this study. We also
note that our method accounts for some of the differences in the simulated MAP by applying the drought metrics
separately to each model's own climatology (such that all drought metrics are calculated relative to the model's
own mean).*

*To clarify this, we have added the following statement at Line 166:* "Note that we do not use the results of this
verification to bias-correct the models. Vogel et al. (2023) used CMIP5 simulations of Australian precipitation to

demonstrate that different bias correction methods can lead to large differences in simulated rainfall climatology, variability and extremes, and that no single bias correction method outperforms the others when evaluated for multiple metrics."

*Vogel, E., F. Johnson, L. Marshall, U. Bende-Michl, L. Wilson, J. R. Peter, C. Wasko, S. Srikanthan, W. Sharples, A. Dowdy, P. Hope, Z. Khan, R. Mehrotra, A. Sharma, V. Matic, A. Oke, M. Turner, S. Thomas, C. Donnelly, V. C. Duong, An evaluation framework for downscaling and bias correction in climate change impact studies, Journal of Hydrology, Volume 622, Part A, 2023. https://doi.org/10.1016/j.jhydrol.2023.129693*

From Sect. 2.3.2, it is inferred that droughts are defined as deviations from the climatology of each model (right?) If the models are bias corrected, the same climatological mean (that from AWAP) could be used for drought definition. And direct comparison between models could be applied, instead of %. This is easier for interpretation than "For example, 0% represents the climatological mean precipitation, and 100% represents zero precipitation". Same for severity, it would be much easier to compare directly mm across models, instead of % ("For example, a value of 200% represents a total deficit equal to two years of mean precipitation.")

Comparing deviation metrics as % is influenced by the native MAP of each model (deviations from a low absolute MAP values represent larger % than when the MAP is larger). By comparing Fig 2.a and Fig. 3, it can be seen that some models have MAP biases up to 100%, with similar absolute biases that observed MAP in Fig. 2a.

**Author response:** *In the case of our analysis, all drought metrics are calculated relative to the models' own climatologies. Therefore, correcting for MAP biases would have a minimal impact on the results as most of the drought metrics presented are based on years above/below the models' own climatological means. The Reviewer is correct that bias-correcting the mean would allow a direct comparison in mm but the relative intensity metric used here in fact already allows for direct comparison across models. Changing this to mm would not change the relative differences across models, and hence would not affect our results or conclusions.*

*Cannon, A.J. Multivariate quantile mapping bias correction: an N-dimensional probability density function transform for climate model simulations of multiple variables. Clim Dyn50, 31–49 (2018). https://doi.org/10.1007/s00382-017-3580-6*

2) Having a more extreme event in a large sample can be somehow expected, but I am missing an assessment of the return period of such events. The longest droughts simulated over the pre-industrial millennium, can be expected to happen over the next century, couple of centuries, thousand years?

**Author response:** *Thank you for this excellent suggestion. We have added a new Supporting Figure (Supp. Fig. 21), showing the return period of multi-year droughts in the Murray-Darling Basin, in each of the PMIP3 models' piLM simulations. In most models, the longest drought occurs only once, giving an estimated return period of around 1000 years (the length of the simulations). However, most models still simulate very long MDB droughts (10-20 years) with relatively short return periods of ~100-150 years. We have added a brief discussion of these results in Section 3.3 (a new subsection 3.3.1, L399) and 4.2 (L502).*

In the same line, I think that for providing evidence for adaptation planning, the longest droughts should be assessed in conjunction with their deficits: it is not the same to communicate that 20-years of minor droughts (e.g., 0-10% deficits) can be expected that to communicate that 20-years of severe droughts (e.g., >40% deficits) can be expected in the future. This could be done by accounting for relative severity together with maximum length.

*Author response:* *Given this adaptation-focussed analysis is most likely to be useful for the Murray-Darling Basin, we have added a new Supporting Figure (Supp. Fig. 20) showing the relative severity associated with the longest drought. On the new Supporting Figure 20, tile colour corresponds to the relative severity of the longest drought, and the text annotation states the length of that drought, in years. As well as the necessary additions to the Methods (L237) and Results (L382), we have added the following text to the Discussion (L505):* "There is low model agreement on the overall severity of the single longest drought occurring in the MDB during the long pre-industrial last millennium and piControl simulations (Supp. Fig. 20). This may in part be an artefact of the 2S2E drought identification method, which allows single wet years during the drought. Hence, droughts can be very long, but not necessarily severe. The nature and drivers of the longest theoretical droughts occurring in the MDB should therefore be investigated in more detail."

**Minor comments:**
Supp. Fig. 7: "Mean multi-year drought length in (a) observations (1900-2000) and (b-l) model simulations of the pre-industrial last millennium (850-1849). Showing the CESM LME ensemble mean." It should say, panel Fig. 7l presents the CESM LME ensemble mean. Same for all figures.
*Author response:* *Thank you for catching this. We have modified the captions for Supporting Figs 1–18 to clarify this.*

Title: it is a complicated title that I don't think is communicating the main messages of the paper. I recommend the authors to consider a simpler one.
*Author response:* *We agree that it is quite a long title(!). Taking into account your suggestions, as well as those of the Editor, we have re-titled the paper* '**Potential for historically unprecedented Australian droughts from natural variability and climate change**'.

Abstract: "Model simulations suggest future droughts across Australia could be much longer than what has been experienced in the twentieth century, even without any human influence." This can be misunderstood as future projections, please re-phrase. An option could be: Drought simulations over the last millennium suggests that future droughts across Australia could be much longer than what has been experienced in the twentieth century, even without any human influence.
*Author response:* *We have modified the sentence to* "Model simulations of droughts over the past millennium suggest future droughts across Australia could be much longer than what has been experienced in the twentieth century, even without any human influence".

---

## Referee Report (RR1)

Review on **Emerging anthropogenic influence on Australian multi-year droughts with potential for historically unprecedented megadroughts**

I thank the authors for carefully addressing my comments and those provided by the other referee. I think the manuscript has improved, the methodology is better explained and the scientific findings are more clearly communicated. I do not have any further comments or suggestions. From my side, it is suitable for publication.